# DiracDiffusion: Denoising and Incremental Reconstruction with Assured Data-Consistency

## Abstract

Diffusion models have established new state of the art in a multitude of computer vision tasks, including image restoration. Diffusion-based inverse problem solvers generate reconstructions of exceptional visual quality from heavily corrupted measurements. However, in what is widely known as the perception-distortion trade-off, the price of perceptually appealing reconstructions is often paid in declined distortion metrics, such as PSNR. Distortion metrics measure faithfulness to the observation, a crucial requirement in inverse problems. In this work, we propose a novel framework for inverse problem solving, namely we assume that the observation comes from a stochastic degradation process that gradually degrades and noises the original clean image. We learn to reverse the degradation process in order to recover the clean image. Our technique maintains consistency with the original measurement throughout the reverse process, and allows for great flexibility in trading off perceptual quality for improved distortion metrics and sampling speedup via early-stopping. We demonstrate the efficiency of our method on different high-resolution datasets and inverse problems, achieving great improvements over other state-of-the-art diffusion-based methods with respect to both perceptual and distortion metrics.

## 1 Introduction

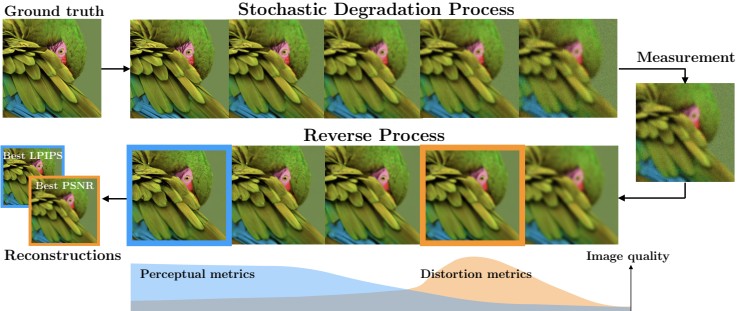

Figure 1: **Overview of our method:** measurement acquisition is modeled as a gradual degradation and noising of an underlying clean ground truth signal via a Stochastic Degradation Process. We reconstruct the clean image from noisy measurements by learning to reverse the degradation process. Our technique allows for obtaining a variety of reconstructions with different perceptual quality-distortion trade-offs, all in a single reverse diffusion sampling trajectory.

Diffusion models (DMs) are powerful generative models capable of synthesizing samples of exceptional quality by reversing a diffusion process that gradually corrupts a clean image by adding Gaussian noise. DMs have been explored from two perspectives: Denoising Diffusion Probabilistic Models (DDPM) Sohl-Dickstein et al. (2015); Ho et al. (2020) and Score-Based Models Song & Ermon (2020a;b), which have been unified under a general framework of Stochastic Differential Equations (SDEs) Song et al. (2020). DMs have established new state of the art in image generation Dhariwal & Nichol (2021); Saharia et al. (2022); Ramesh et al. (2022); Rombach et al. (2022),

audio Kong et al. (2020) and video synthesis Ho et al. (2022). Recently, there has been a push to broaden the notion of Gaussian diffusion, such as extension to other noise distributions Deasy et al. (2021); Nachmani et al. (2021); Okhotin et al. (2023). In the context of image generation, there has been work to generalize the corruption process, such as blur diffusion Lee et al. (2022); Hoogeboom & Salimans (2022), inverse heat dissipation Rissanen et al. (2022) and arbitrary linear corruptions Daras et al. (2022) with Bansal et al. (2022) questioning the necessity of stochasticity in the generative process all together. However, designing the diffusion process specifically for inverse problem solving has not been extensively explored yet.

In inverse problems, one wishes to recover a signal $x$ from a noisy observation $y = \mathcal{A}(x) + z$ where $\mathcal{A}$ is typically non-invertible. The unconditional score function learned by DMs has been successfully leveraged to solve inverse problems without any task-specific training Kadkhodaie & Simoncelli (2021); Jalal et al. (2021); Saharia et al. (2021) resulting in reconstructions with exceptional perceptual quality. However, these methods underperform in distortion metrics, such as PSNR and SSIM Chung et al. (2022a) due to the so called perception-distortion trade-off Blau & Michaeli (2018). Authors in Delbracio & Milanfar (2023) observe that in their framework, the total number of restoration steps controls the perception-distortion trade-off, with less steps yielding results closer to the minimum distortion estimate. Similar observation is made in Whang et al. (2022) in the context of blind image deblurring, where authors additionally propose to average multiple reconstructions for improved distortion metrics. Authors in Kawar et al. (2022a) report that, the amount of noise injected at each timestep controls the trade-off between reconstruction error and image quality.

Beyond image quality, a key requirement imposed on reconstructions is data consistency, that is faithfulness to the original observation. In the context of diffusion-based solvers, different methods have been proposed to enforce consistency between the generated image and the corresponding observations. These methods include alternating between a step of unconditional update and a step of projection Song et al. (2021b); Chung & Ye (2022); Chung et al. (2022c) or other correction techniques Chung et al. (2022a;b) to guide the diffusion process towards data consistency. Another line of work proposes diffusion in the spectral space of the forward operator, achieving high quality reconstructions, however requires costly singular value decomposition Kawar et al. (2021; 2022a;b). Song et al. (2023) uses pseudo-inverse guidance to incorporate the model into the reconstruction process. All of these methods utilize a pre-trained score function learned for a standard diffusion process that simply adds Gaussian noise to clean images. Recently, there has been some work on extending Gaussian diffusion by incorporating the image degradation into the score-model training procedure. A recent example is Welker et al. (2022) proposing adding an additional drift term to the forward SDE that pulls the iterates towards the corrupted measurement and demonstrates high quality reconstructions for JPEG compression artifact removal. A blending parametrization Heitz et al. (2023); Delbracio & Milanfar (2023) has been proposed that defines the forward process as convex combinations between the clean image and corrupted observation. Liu et al. (2023) leverages Schrödinger bridges for image restoration, a nonlinear extension of score-based models defined between degraded and clean image distributions. Even though these methods utilize degraded-clean image pairs for training, they don't explicitly leverage the forward operator for score-model training.

In this paper, we propose a novel framework for solving inverse problems using a generalized notion of diffusion that mimics the corruption process that produced the observation. We call our method *Dirac*: Denoising and Incremental Reconstruction with Assured data-Consistency. As the forward model and noising process are directly incorporated into the framework, our method maintains data consistency throughout the reverse diffusion process, without any additional steps such as projections. Furthermore, we make the key observation that details are gradually added to the posterior mean estimates during the sampling process. This property imbues our method with great flexibility: by leveraging early-stopping we can freely trade off perceptual quality for better distortion metrics and sampling speedup or vice versa. We provide theoretical analysis on the accuracy and limitations of our method that are well-supported by empirical results. Our numerical experiments demonstrate state-of-the-art results in terms of both perceptual and distortion metrics with fast sampling.

## 2 BACKGROUND

**Diffusion models –** DMs are generative models based on a corruption process that gradually transforms a clean image distribution $q_0$ into a known prior distribution which is tractable, but contains no information of data. The corruption level, or *severity* as we refer to it in this paper, is indexed by time $t$ and increases from $t = 0$ (clean images) to $t = 1$ (pure noise). The typical corrup-

tion process consists of adding Gaussian noise of increasing magnitude to clean images, that is $q_t(\boldsymbol{x}_t|\boldsymbol{x}_0) \sim \mathcal{N}(\boldsymbol{x}_0, \sigma_t^2 \mathbf{I})$, where $\boldsymbol{x}_0 \sim q_0$ is a clean image, and $\boldsymbol{x}_t$ is the corrupted image at time $t$. By learning to reverse the corruption process, one can generate samples from $q_0$ by sampling from a simple noise distribution and running the learned reverse diffusion process from $t = 1$ to $t = 0$.

DMs have been explored along two seemingly different trajectories. Score-Based Models Song & Ermon (2020a;b) attempt to learn the gradient of the log likelihood and use Langevin dynamics for sampling, whereas DDPM Sohl-Dickstein et al. (2015); Ho et al. (2020) adopts a variational inference interpretation. More recently, a unified framework based on SDEs Song et al. (2020) has been proposed. Namely, both Score-Based Models and DDPM can be expressed via a Forward SDE in the form $\mathrm{d}\boldsymbol{x} = f(\boldsymbol{x}, t)\mathrm{d}t + g(t)\mathrm{d}\boldsymbol{w}$ with different choices of $f$ and $g$. Here $\boldsymbol{w}$ denotes the standard Wiener process. This SDE is reversible Anderson (1982), and the Reverse SDE can be written as

$$\mathrm{d}\boldsymbol{x} = [f(\boldsymbol{x}, t) - g^2(t)\nabla_{\boldsymbol{x}} \log q_t(\boldsymbol{x})]\mathrm{d}t + g(t)\mathrm{d}\bar{\boldsymbol{w}}, \tag{1}$$

where $\bar{\boldsymbol{w}}$ is the standard Wiener process, where time flows in the reverse direction. The true score $\nabla_{\boldsymbol{x}} \log q_t(\boldsymbol{x})$ is approximated by a neural network $s_{\boldsymbol{\theta}}(\boldsymbol{x}_t, t)$ from the tractable conditional distribution $q_t(\boldsymbol{x}_t|\boldsymbol{x}_0)$ by minimizing

$$\mathbb{E}_{t \sim U[0,1], (\boldsymbol{x}_0, \boldsymbol{x}_t)} \left[ w(t) \left\| s_{\boldsymbol{\theta}}(\boldsymbol{x}_t, t) - \nabla_{\boldsymbol{x}_t} q_t(\boldsymbol{x}_t|\boldsymbol{x}_0) \right\|^2 \right], \tag{2}$$

where $(\boldsymbol{x}_0, \boldsymbol{x}_t) \sim q_0(\boldsymbol{x}_0) q_t(\boldsymbol{x}_t|\boldsymbol{x}_0)$ and $w(t)$ is a weighting function.

**Diffusion Models for Inverse problems –** Our goal is to solve a noisy inverse problem

$$\tilde{\boldsymbol{y}} = \mathcal{A}(\boldsymbol{x}_0) + \boldsymbol{z}, \ \boldsymbol{z} \sim \mathcal{N}(\mathbf{0}, \sigma^2 \mathbf{I}), \tag{3}$$

with $\tilde{\boldsymbol{y}}, \boldsymbol{x}_0 \in \mathbb{R}^n$ and $\mathcal{A} : \mathbb{R}^n \to \mathbb{R}^n$. That is, we are interested in solving a reconstruction problem, where we observe a measurement $\tilde{\boldsymbol{y}}$ that is known to be produced by applying a non-invertible mapping $\mathcal{A}$ to a ground truth signal $\boldsymbol{x}_0$ and is corrupted by additive noise $\boldsymbol{z}$. We refer to $\mathcal{A}$ as the degradation, and $\mathcal{A}(\boldsymbol{x}_0)$ as a degraded signal. Our goal is to recover $\boldsymbol{x}_0$ as faithfully as possible, which can be thought of as generating samples from the posterior distribution $q(\boldsymbol{x}_0|\tilde{\boldsymbol{y}})$. Diffusion models have emerged as useful priors enabling sampling from the posterior based on (1). Using Bayes rule, the score of the posterior can be written as $\nabla_{\boldsymbol{x}} \log q_t(\boldsymbol{x}|\tilde{\boldsymbol{y}}) = \nabla_{\boldsymbol{x}} \log q_t(\boldsymbol{x}) + \nabla_{\boldsymbol{x}} \log q_t(\tilde{\boldsymbol{y}}|\boldsymbol{x})$, where the first term can be approximated using score-matching as in (2). On the other hand, the second term cannot be expressed in closed-form in general, and therefore a flurry of activity emerged recently to circumvent computing the likelihood directly.

## 3 METHOD

In this work, we propose a novel perspective on solving ill-posed inverse problems, where the forward model is known. In particular, we assume that our noisy observation $\tilde{\boldsymbol{y}}$ results from a process that gradually applies more and more severe degradations to an underlying clean signal.

### 3.1 DEGRADATION SEVERITY

To define severity more rigorously, we appeal to the intuition that given two noiseless, degraded signals $\boldsymbol{y}$ and $\boldsymbol{y}^+$ of a clean signal $\boldsymbol{x}_0$, then $\boldsymbol{y}^+$ is corrupted by a more severe degradation than $\boldsymbol{y}$, if $\boldsymbol{y}$ contains all the information necessary to find $\boldsymbol{y}^+$ without knowing $\boldsymbol{x}_0$.

**Definition 3.1** (Severity of degradations). A mapping $\mathcal{A}_+ : \mathbb{R}^n \to \mathbb{R}^n$ is a *more severe degradation than* $\mathcal{A} : \mathbb{R}^n \to \mathbb{R}^n$ if there exists a surjective mapping $\mathcal{G}_{\mathcal{A} \to \mathcal{A}_+} : Image(\mathcal{A}) \to Image(\mathcal{A}_+)$. That is,

$$\mathcal{A}_+(\boldsymbol{x}_0) = \mathcal{G}_{\mathcal{A} \to \mathcal{A}_+}(\mathcal{A}(\boldsymbol{x}_0)) \ \forall \boldsymbol{x}_0 \in dom(\mathcal{A}).$$

We call $\mathcal{G}_{\mathcal{A} \to \mathcal{A}_+}$ the *forward degradation transition function* from $\mathcal{A}$ to $\mathcal{A}_+$.

Take image inpainting as an example (Fig. 2) and let $\mathcal{A}_t$ denote a masking operator that sets pixels to 0 within a centered box, where the box side length is $l(t) = t \cdot W$, where $W$ is the image width and $t \in [0, 1]$. Assume that we have an observation $\boldsymbol{y}_{t'} = \mathcal{A}_{t'}(\boldsymbol{x}_0)$ which is a degradation of a clean image $\boldsymbol{x}_0$ where a small center square with side length $l(t')$ is masked out. Given $\boldsymbol{y}_{t'}$, without having access to the complete clean image, we can find any other masked version of $\boldsymbol{x}_0$ where a box with at least side length $l(t')$ is masked out. Therefore every other masking operator $\mathcal{A}_{t''}$, $t' < t''$ is a more severe degradation than $\mathcal{A}_{t'}$. The forward degradation transition function $\mathcal{G}_{\mathcal{A}_{t'} \to \mathcal{A}_{t''}}$ in this case is simply $\mathcal{A}_{t''}$. We also note here, that the *reverse degradation transition function* $\mathcal{H}_{\mathcal{A}_{t''} \to \mathcal{A}_{t'}}$ that recovers $\mathcal{A}_{t'}(\boldsymbol{x}_0)$ from a more severe degradation $\mathcal{A}_{t''}(\boldsymbol{x}_0)$ for any $\boldsymbol{x}_0$ does not exist in general.

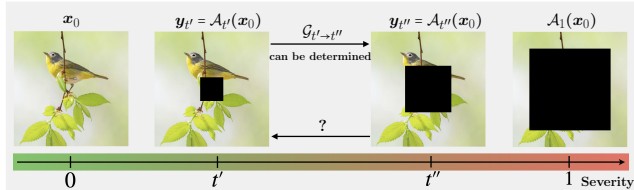

Figure 2: Severity of degradations: We can always find a more degraded image $\boldsymbol{y}_{t''}$ from a less degraded version of the same clean image $\boldsymbol{y}_{t'}$ via the forward degradation transition function $\mathcal{G}_{t' \rightarrow t''}$, but not vice versa.

### 3.2 DETERMINISTIC AND STOCHASTIC DEGRADATION PROCESSES

Using this novel notion of degradation severity, we can define a deterministic degradation process that gradually removes information from the clean signal via more and more severe degradations.

**Definition 3.2** (Deterministic degradation process). A *deterministic degradation process* is a differentiable mapping $\mathcal{A} : [0, 1] \times \mathbb{R}^n \rightarrow \mathbb{R}^n$ that has the following properties:

1. *Diminishing severity:* $\mathcal{A}(0, \boldsymbol{x}) = \boldsymbol{x}$
2. *Monotonically degrading:* $\forall t' \in [0, 1)$ and $t'' \in (t', 1]$ $\mathcal{A}(t'', \cdot)$ is a more severe degradation than $\mathcal{A}(t', \cdot)$.

We use the shorthand $\mathcal{A}(t, \cdot) = \mathcal{A}_t(\cdot)$ and $\mathcal{G}_{\mathcal{A}_{t'} \rightarrow \mathcal{A}_{t''}} = \mathcal{G}_{t' \rightarrow t''}$ for the underlying forward degradation transition functions for all $t' < t''$. Our deterministic degradation process starts from a clean signal $\boldsymbol{x}_0$ at time $t = 0$ and applies degradations with increasing severity over time. If we choose $\mathcal{A}(1, \cdot) = \boldsymbol{0}$, then all information in the original signal is destroyed over the degradation process. One can sample easily from the *forward process*, that is the process that evolves forward in time, starting from a clean image $\boldsymbol{x}_0$ at $t = 0$. A sample from time $t$ can be computed directly as $\boldsymbol{y}_t = \mathcal{A}_t(\boldsymbol{x}_0)$.

In order to account for measurement noise, one can combine the deterministic degradation process with a stochastic noising process that gradually adds Gaussian noise to the degraded measurements.

**Definition 3.3** (Stochastic degradation process (SDP)). $\boldsymbol{y}_t = \mathcal{A}_t(\boldsymbol{x}_0) + \boldsymbol{z}_t$, $\boldsymbol{z}_t \sim \mathcal{N}(\boldsymbol{0}, \sigma_t^2 \mathbf{I})$ is a *stochastic degradation process* if $\mathcal{A}_t$ is a deterministic degradation process, $t \in [0, 1]$, and $\boldsymbol{x}_0 \sim q_0(\boldsymbol{x}_0)$ is a sample from the clean data distribution. We denote the distribution of $\boldsymbol{y}_t$ as $q_t(\boldsymbol{y}_t) \sim \mathcal{N}(\mathcal{A}_t(\boldsymbol{x}_0), \sigma_t^2 \mathbf{I})$.

A key contribution of our work is looking at a noisy, degraded signal as a sample from the forward process of an underlying SDP, and considering the reconstruction problem as running the reverse process of the SDP backwards in time in order to recover the clean sample. Our formulation interpolates between degraded and clean image distributions through a severity parametrization that requires an analytical form of $\mathcal{A}(\cdot)$. An alternative approach Delbracio & Milanfar (2023); Heitz et al. (2023) is to parametrize intermediate distributions as convex combinations of corresponding pairs of noisy and clean samples as $\boldsymbol{y}_t = t\tilde{\boldsymbol{y}} + (1 - t)\boldsymbol{x}_0$, $t \in [0, 1]$, also referred to as *blending* Heitz et al. (2023). In our framework, this formulation can be thought of as a deterministic degradation process $\mathcal{A}_t(\boldsymbol{x}_0; \tilde{\boldsymbol{y}}) = t\tilde{\boldsymbol{y}} + (1 - t)\boldsymbol{x}_0$ conditioned on $\tilde{\boldsymbol{y}}$. However, as the underlying degradation operator is not leveraged in this formulation, we cannot develop theoretical guarantees on data consistency of the reconstruction. Moreover, we observe improved noise robustness using the proposed SDP formulation. For a more detailed comparison we refer the reader to Appendix G.

### 3.3 SDP AS A STOCHASTIC DIFFERENTIAL EQUATION

We can formulate the evolution of our degraded and noisy measurements $\boldsymbol{y}_t$ as an SDE:

$$\mathrm{d}\boldsymbol{y}_t = \dot{\mathcal{A}}_t(\boldsymbol{x}_0)\mathrm{d}t + \sqrt{\frac{\mathrm{d}}{\mathrm{d}t}\sigma_t^2}\mathrm{d}w. \tag{4}$$

This is an example of an Itô-SDE, and for a fixed $\boldsymbol{x}_0$ the above process is reversible, where the reverse diffusion process is given by

$$\mathrm{d}\boldsymbol{y}_t = \left(\dot{\mathcal{A}}_t(\boldsymbol{x}_0)\mathrm{d}t - \left(\frac{\mathrm{d}}{\mathrm{d}t}\sigma_t^2\right)\nabla_{\boldsymbol{y}_t}\log q_t(\boldsymbol{y}_t)\right)\mathrm{d}t + \sqrt{\frac{\mathrm{d}}{\mathrm{d}t}\sigma_t^2}\mathrm{d}\bar{w}. \tag{5}$$

One would solve the above SDE by discretizing it (for example Euler-Maruyama), approximating differentials with finite differences:

$$\boldsymbol{y}_{t-\Delta t} = \boldsymbol{y}_t + \underbrace{\mathcal{A}_{t-\Delta t}(\boldsymbol{x}_0) - \mathcal{A}_t(\boldsymbol{x}_0)}_{\text{incremental reconstruction}} - \underbrace{(\sigma_{t-\Delta_t}^2 - \sigma_t^2)\nabla_{\boldsymbol{y}_t}\log q_t(\boldsymbol{y}_t)}_{\text{denoising}} + \sqrt{\sigma_t^2 - \sigma_{t-\Delta_t}^2}\boldsymbol{z}, \quad (6)$$

where $\boldsymbol{z} \sim \mathcal{N}(\boldsymbol{0}, \mathbf{I})$. The update in (6) lends itself to an interesting interpretation. One can look at it as the combination of a small, incremental reconstruction and denoising steps. In particular, assume that $\boldsymbol{y}_t = \mathcal{A}_t(\boldsymbol{x}_0) + \boldsymbol{z}_t$ and let

$$\mathcal{R}(t, \Delta t; \boldsymbol{x}_0) := \mathcal{A}_{t-\Delta t}(\boldsymbol{x}_0) - \mathcal{A}_t(\boldsymbol{x}_0). \quad (7)$$

Then, the first term $\boldsymbol{y}_t + \mathcal{R}(t, \Delta t; \boldsymbol{x}_0) = \mathcal{A}_{t-\Delta t}(\boldsymbol{x}_0) + \boldsymbol{z}_t$ will reverse a $\Delta t$ step of the deterministic degradation process, equivalent in effect to the reverse degradation transition function $\mathcal{H}_{t \to t-\Delta t}$. The second term is analogous to a denoising step in standard diffusion, where a slightly less noisy version of the image is predicted. However, before we can simulate the reverse SDE in (6) to recover $\boldsymbol{x}_0$, we face two obstacles. First, we do not know the score of $q_t(\boldsymbol{y}_t)$. This is commonly tackled by learning a noise-conditioned score network that matches $\log q_t(\boldsymbol{y}_t|\boldsymbol{x}_0)$ which we can easily compute. We are also going to follow this path. Second, we do not know $\mathcal{A}_{t-\Delta t}(\boldsymbol{x}_0)$ and $\mathcal{A}_t(\boldsymbol{x}_0)$ for the incremental reconstruction step, since $\boldsymbol{x}_0$ is unknown to us when reversing the degradation process.

### 3.4 DENOISING - LEARNING A SCORE NETWORK

To run the reverse SDE, we need the score of the noisy, degraded distribution $\nabla_{\boldsymbol{y}_t}\log q_t(\boldsymbol{y}_t)$, which is intractable. However, we can use the denoising score matching framework to approximate the score. In particular, instead of the true score, we can easily compute the score for the conditional distribution, when the clean image $\boldsymbol{x}_0$ is given as $\nabla_{\boldsymbol{y}_t}\log q_t(\boldsymbol{y}_t|\boldsymbol{x}_0) = \frac{\mathcal{A}_t(\boldsymbol{x}_0)-\boldsymbol{y}_t}{\sigma_t^2}$. During training, we have access to clean images $\boldsymbol{x}_0$ and can generate any degraded, noisy image $\boldsymbol{y}_t$ using our SDP formulation $\boldsymbol{y}_t = \mathcal{A}_t(\boldsymbol{x}_0) + \boldsymbol{z}_t$. Thus, we learn an estimator of the conditional score function $s_{\boldsymbol{\theta}}(\boldsymbol{y}_t, t)$ by minimizing

$$\mathcal{L}_t(\boldsymbol{\theta}) = \mathbb{E}_{(\boldsymbol{x}_0, \boldsymbol{y}_t)}\left[\left\|s_{\boldsymbol{\theta}}(\boldsymbol{y}_t, t) - \frac{\mathcal{A}_t(\boldsymbol{x}_0) - \boldsymbol{y}_t}{\sigma_t^2}\right\|^2\right], \quad (8)$$

where $(\boldsymbol{x}_0, \boldsymbol{y}_t) \sim q_0(\boldsymbol{x}_0)q_t(\boldsymbol{y}_t|\boldsymbol{x}_0)$. One can show that the well-known result of Vincent (2011) applies to our SDP formulation, and thus by minimizing the objective in (8), we can learn the score $\nabla_{\boldsymbol{y}_t}\log q_t(\boldsymbol{y}_t)$ (see details in Appendix A.1).

We parameterize the score network as

$$s_{\boldsymbol{\theta}}(\boldsymbol{y}_t, t) = \frac{\mathcal{A}_t(\Phi_{\boldsymbol{\theta}}(\boldsymbol{y}_t, t)) - \boldsymbol{y}_t}{\sigma_t^2}, \quad (9)$$

that is given a noisy and degraded image as input, the model predicts the underlying clean image $\boldsymbol{x}_0$. Other parametrizations are also possible, such as predicting $\boldsymbol{z}_t$ or (equivalently) predicting $\mathcal{A}_t(\boldsymbol{x}_0)$. However, as pointed out in Daras et al. (2022), this might lead to learning the image distribution only locally, around degraded images. Furthermore, in order to estimate the incremental reconstruction $\mathcal{R}(t, \Delta t; \boldsymbol{x}_0)$, we not only need to estimate $\mathcal{A}_t(\boldsymbol{x}_0)$, but other functions of $\boldsymbol{x}_0$, and thus estimating $\boldsymbol{x}_0$ directly gives us more flexibility. Rewriting (8) with the new parametrization leads to

$$\mathcal{L}(\boldsymbol{\theta}) = \mathbb{E}_{t, (\boldsymbol{x}_0, \boldsymbol{y}_t)}\left[w(t)\|\mathcal{A}_t(\Phi_{\boldsymbol{\theta}}(\boldsymbol{y}_t, t)) - \mathcal{A}_t(\boldsymbol{x}_0)\|^2\right], \quad (10)$$

where $t \sim U[0, 1]$, $(\boldsymbol{x}_0, \boldsymbol{y}_t) \sim q_0(\boldsymbol{x}_0)q_t(\boldsymbol{y}_t|\boldsymbol{x}_0)$ and typical choices in the diffusion literature for the weights $w(t)$ are 1 or $1/\sigma_t^2$. Intuitively, the neural network receives a noisy, degraded image, along with the degradation severity, and outputs a prediction $\hat{\boldsymbol{x}}_0(\boldsymbol{y}_t) = \Phi_{\boldsymbol{\theta}}(\boldsymbol{y}_t, t)$ such that the *degraded* ground truth $\mathcal{A}_t(\boldsymbol{x}_0)$ and the *degraded* prediction $\mathcal{A}_t(\hat{\boldsymbol{x}}_0(\boldsymbol{y}_t))$ are consistent.

### 3.5 INCREMENTAL RECONSTRUCTIONS

Given an estimator of the score, we still need to approximate $\mathcal{R}(t, \Delta t; \boldsymbol{x}_0)$ in order to run the reverse SDE in (6). That is we have to estimate *how the degraded image changes if we slightly decrease the degradation severity*. As we parameterize our score network in (9) to learn a representation of the clean image manifold directly, we can estimate the incremental reconstruction term as

$$\hat{\mathcal{R}}(t, \Delta t; \boldsymbol{y}_t) = \mathcal{A}_{t-\Delta t}(\Phi_{\boldsymbol{\theta}}(\boldsymbol{y}_t, t)) - \mathcal{A}_t(\Phi_{\boldsymbol{\theta}}(\boldsymbol{y}_t, t)). \quad (11)$$

One may consider this a *look-ahead method*, since we use $\boldsymbol{y}_t$ with degradation severity $t$ to predict a less severe degradation of the clean image "ahead" in the reverse process. This becomes more obvious when we note, that our score network already learns to predict $\mathcal{A}_t(\boldsymbol{x}_0)$ given $\boldsymbol{y}_t$ due to the training loss in (10). However, even if we learn the true score perfectly via (10), there is no guarantee that $\mathcal{A}_{t-\Delta t}(\boldsymbol{x}_0) \approx \mathcal{A}_{t-\Delta t}(\Phi_{\boldsymbol{\theta}}(\boldsymbol{y}_t, t))$. The following result provides an upper bound on the approximation error.

**Theorem 3.4.** *Let $\hat{\mathcal{R}}(t, \Delta t; \boldsymbol{y}_t)$ from (11) denote our estimate of the incremental reconstruction, where $\Phi_{\boldsymbol{\theta}}(\boldsymbol{y}_t, t)$ is trained on the loss in (10). Let $\mathcal{R}^*(t, \Delta t; \boldsymbol{y}_t) = \mathbb{E}[\mathcal{R}(t, \Delta t; \boldsymbol{x}_0)|\boldsymbol{y}_t]$ denote the MMSE estimator of $\mathcal{R}(t, \Delta t; \boldsymbol{x}_0)$. Assume, that the degradation process is smooth such that $\|\mathcal{A}_t(\boldsymbol{x}) - \mathcal{A}_t(\boldsymbol{x}')\| \leq L_x^{(t)} \|\boldsymbol{x} - \boldsymbol{x}'\|$, $\forall \boldsymbol{x}, \boldsymbol{x}' \in \mathbb{R}^n$ and $\|\mathcal{A}_t(\boldsymbol{x}) - \mathcal{A}_{t'}(\boldsymbol{x})\| \leq L_t|t - t'|$, $\forall t, t' \in [0, 1]$, $\forall \boldsymbol{x} \in \mathbb{R}^n$. Further assume that the clean images have bounded entries $\boldsymbol{x}_0[i] \leq B$, $\forall i \in (1, 2, ..., n)$ and that the error in our score network is bounded by $\|s_{\boldsymbol{\theta}}(\boldsymbol{y}_t, t) - \nabla_{\boldsymbol{y}_t} \log q_t(\boldsymbol{y}_t)\| \leq \frac{\epsilon_t}{\sigma_t^2}$, $\forall t \in [0, 1]$. Then,*

$$\|\hat{\mathcal{R}}(t, \Delta t; \boldsymbol{y}_t) - \mathcal{R}^*(t, \Delta t; \boldsymbol{y}_t)\| \leq \underbrace{(L_x^{(t)} + L_x^{(t-\Delta t)})}_{\text{degr. smoothness}} \underbrace{\sqrt{n}B}_{\text{data}} + \underbrace{2L_t}_{\text{scheduling}} \underbrace{\Delta t}_{\text{algorithm}} + \underbrace{2\epsilon_t}_{\text{optimization}} .$$

The first term in the upper bound suggests that smoother degradations are easier to reconstruct accurately. The second term indicates two crucial points: (1) sharp variations in the degradation with respect to time leads to potentially large estimation error and (2) the error can be controlled by choosing a small enough step size in the reverse process. Scheduling of the degradation over time is a design parameter, and Theorem 3.4 suggests that sharp changes with respect to $t$ should be avoided. Finally, the error grows with less accurate score estimation, however with large enough network capacity, this term can be driven close to 0.

The main contributor to the error in Theorem 3.4 stems from the fact that consistency under less severe degradations, that is $\mathcal{A}_{t-\Delta t}(\Phi_{\boldsymbol{\theta}}(\boldsymbol{y}_t, t)) \approx \mathcal{A}_{t-\Delta t}(\boldsymbol{x}_0)$, is not enforced by the loss in (10). To this end, we propose a novel loss function, the *incremental reconstruction loss*, that combines learning to denoise and reconstruct simultaneously:

$$\mathcal{L}_{IR}(\Delta t, \boldsymbol{\theta}) = \mathbb{E}_{t, (\boldsymbol{x}_0, \boldsymbol{y}_t)} \left[ w(t) \|\mathcal{A}_\tau(\Phi_{\boldsymbol{\theta}}(\boldsymbol{y}_t, t)) - \mathcal{A}_\tau(\boldsymbol{x}_0)\|^2 \right], \tag{12}$$

where $\tau = \max(t - \Delta t, 0)$, $t \sim U[0, 1]$, $(\boldsymbol{x}_0, \boldsymbol{y}_t) \sim q_0(\boldsymbol{x}_0)q_t(\boldsymbol{y}_t|\boldsymbol{x}_0)$. It is clear, that minimizing this loss directly improves our estimate of the incremental reconstruction in (11). We find that if $\Phi_{\boldsymbol{\theta}}$ has large enough capacity, minimizing the incremental reconstruction loss in (12) also implies minimizing (10), and thus the true score is learned (denoising is achieved). Furthermore, we show that (12) is an upper bound to (10) (Appendix A.3). By minimizing (12), the model learns not only to denoise, but also to perform small, incremental reconstructions of the degraded image such that $\mathcal{A}_{t-\Delta t}(\Phi_{\boldsymbol{\theta}}(\boldsymbol{y}_t, t)) \approx \mathcal{A}_{t-\Delta t}(\boldsymbol{x}_0)$. There is however a trade-off between incremental reconstruction performance and learning the score: we are optimizing an upper bound to (10) and thus it is possible that the score estimation is less accurate. We expect incremental reconstruction loss to work best in scenarios where the degradation may change rapidly with respect to $t$ and hence a network trained to estimate $\mathcal{A}_t(\boldsymbol{x}_0)$ from $\boldsymbol{y}_t$ may become inaccurate when predicting $\mathcal{A}_{t-\Delta t}(\boldsymbol{x}_0)$ from $\boldsymbol{y}_t$.

## 3.6 DATA CONSISTENCY

Data consistency is a crucial requirement on generated images when solving inverse problems. That is, we want to obtain reconstructions that are consistent with our original measurement under the degradation model. More formally, we define data consistency as follows in our framework.

**Definition 3.5** (Data consistency). Given a deterministic degradation process $\mathcal{A}_t(\cdot)$, two degradation severities $\tau \in [0, 1]$ and $\tau^+ \in [\tau, 1]$ and corresponding degraded images $\boldsymbol{y}_\tau \in \mathbb{R}^n$ and $\boldsymbol{y}_{\tau^+} \in \mathbb{R}^n$, $\boldsymbol{y}_{\tau^+}$ is *data consistent* with $\boldsymbol{y}_\tau$ under $\mathcal{A}_t(\cdot)$ if $\exists \boldsymbol{x}_0 \in \mathcal{X}_0$ such that $\mathcal{A}_\tau(\boldsymbol{x}_0) = \boldsymbol{y}_\tau$ and $\mathcal{A}_{\tau^+}(\boldsymbol{x}_0) = \boldsymbol{y}_{\tau^+}$, where $\mathcal{X}_0$ denotes the clean image manifold. We use the notation $\boldsymbol{y}_{\tau^+} \overset{d.c.}{\sim} \boldsymbol{y}_\tau$.

Simply put, two degraded images are data consistent, if there is a clean image which may explain both under the deterministic degradation process. As our proposed technique is directly trained to reverse a degradation process, enforcement of data consistency is built-in without applying additional steps, such as projection. The following theorem guarantees that in the ideal case, data consistency is maintained in *each iteration* of the reconstruction algorithm. Proof is provided in Appendix A.4.

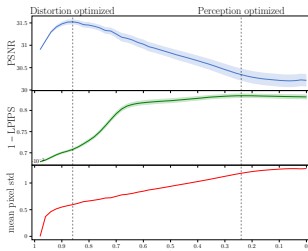 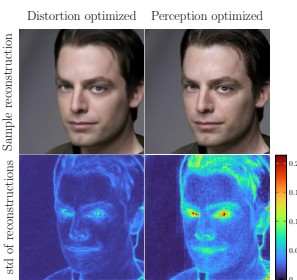

Figure 3: Perception-distortion trade-off on CelebA-HQ deblurring: distortion metrics initially improve, peak fairly early in the reverse process, then gradually deteriorate, while perceptual metrics improve. We plot the mean of 30 trajectories ($\pm std$ shaded) starting from the same measurement.

**Theorem 3.6** (Data consistency over iterations). *Assume that we run the updates in* (6) *with* $s_{\boldsymbol{\theta}}(\boldsymbol{y}_t, t) = \nabla_{\boldsymbol{y}_t} \log q_t(\boldsymbol{y}_t)$, $\forall t \in [0, 1]$ *and* $\hat{\mathcal{R}}(t, \Delta t; \boldsymbol{y}_t) = \mathcal{R}(t, \Delta t; \boldsymbol{x}_0)$, $\boldsymbol{x}_0 \in \mathcal{X}_0$. *If we start from a noisy degraded observation* $\tilde{\boldsymbol{y}} = \mathcal{A}_1(\boldsymbol{x}_0) + \boldsymbol{z}_1$, $\boldsymbol{x}_0 \in \mathcal{X}_0$, $\boldsymbol{z}_1 \sim \mathcal{N}(\boldsymbol{0}, \sigma_1^2 \boldsymbol{I})$ *and run the updates in* (6) *for* $\tau = 1, 1 - \Delta t, ..., \Delta t, 0$, *then*

$$\mathbb{E}[\tilde{\boldsymbol{y}}] \overset{d.c.}{\sim} \mathbb{E}[\boldsymbol{y}_\tau], \; \forall \tau \in [1, 1 - \Delta t, ..., \Delta t, 0]. \tag{13}$$

### 3.7 Perception-distortion trade-off

Diffusion models generate synthetic images of exceptional quality, almost indistinguishable from real images to the human eye. This perceptual image quality is typically evaluated on features extracted by a pre-trained neural network, resulting in metrics such as Learned Perceptual Image Patch Similarity (LPIPS)Zhang et al. (2018) or Fréchet Inception Distance (FID)Heusel et al. (2017). In image restoration however, we are often interested in image distortion metrics that reflect faithfulness to the original image, such as Peak Signal to Noise Ratio (PSNR) or Structural Similarity Index Measure (SSIM) when evaluating the quality of reconstructions. Interestingly, distortion and perceptual quality are fundamentally at odds with each other, as shown in the seminal work of Blau & Michaeli (2018). As diffusion models tend to favor high perceptual quality, it is often at the detriment of distortion metrics Chung et al. (2022a).

As shown in Figure 3, we empirically observe that in the reverse process of *Dirac*, the quality of reconstructions with respect to distortion metrics initially improves, peaks fairly early in the reverse process, then gradually deteriorates. Simultaneously, perceptual metrics such as LPIPS demonstrate stable improvement for most of the reverse process. More intuitively, the algorithm first finds a rough reconstruction that is consistent with the measurement, but lacks fine details. This reconstruction is optimal with respect to distortion metrics, but visually overly smooth and blurry. Consecutively, image details progressively emerge during the rest of the reverse process, resulting in improving perceptual quality at the cost of deteriorating distortion metrics. Therefore, our method provides an additional layer of flexibility: by *early-stopping* the reverse process, we can trade-off perceptual quality for better distortion metrics. Adjusting the early-stopping parameter $t_{stop}$ allows us to obtain distortion- and perception-optimized reconstructions depending on our requirements.

### 3.8 Degradation scheduling

In order to deploy our method, we need to define how the degradation changes with respect to severity $t$ following the properties specified in Definition 3.3. That is, we have to determine how to interpolate between the identity mapping $\mathcal{A}_0(\boldsymbol{x}) = \boldsymbol{x}$ for $t = 0$ and the most severe degradation $\mathcal{A}_1(\cdot)$ for $t = 1$. Theorem 3.4 suggests that sharp changes in the degradation function with respect to $t$ should be avoided, however we propose a more principled method of scheduling. In particular, we use a greedy algorithm to select a set of degraded distributions, such that the maximum distance between them is minimized. We define the distance between distributions as $\mathbb{E}_{\boldsymbol{x}_0 \sim \mathcal{X}_0}[\mathcal{M}(\mathcal{A}_i(\boldsymbol{x}_0), \mathcal{A}_j(\boldsymbol{x}_0))]$, where $\mathcal{M}$ is a pairwise image dissimilarity metric. Details can be found in Appendix D.

## 4 Experiments

**Experimental setup –** We evaluate our method on CelebA-HQ ($256 \times 256$) Karras et al. (2018) and ImageNet ($256 \times 256$) Deng et al. (2009). For competing methods that require a score model, we use pre-trained SDE-VP models. For *Dirac*, we train models from scratch using the NCSN++Song

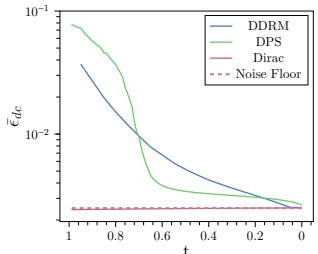 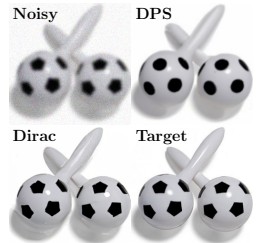 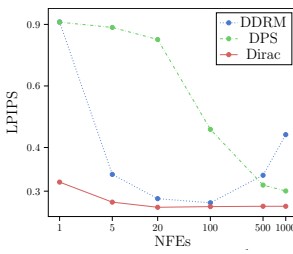

Figure 4: Left: Data consistency in FFHQ inpainting. $\epsilon_{dc} := \|\tilde{\boldsymbol{y}} - \mathcal{A}_1(\hat{\boldsymbol{x}}_0(\boldsymbol{y}_t))\|^2$ measures how consistent is the clean image estimate with the measurement. We expect $\epsilon_{dc}$ to approach the noise floor $\sigma_1^2 = 0.0025$ in case of perfect data consistency. We plot $\bar{\epsilon}_{dc}$ the mean over the validation set. *Dirac* maintains data consistency throughout the reverse process. Center: Data consistency is not always achieved with DPS. Right: Number of reverse diffusion steps vs. perceptual quality.

| Method | Deblurring | | | | Inpainting | | | |
|---|---|---|---|---|---|---|---|---|
| | PSNR($\uparrow$) | SSIM($\uparrow$) | LPIPS($\downarrow$) | FID($\downarrow$) | PSNR($\uparrow$) | SSIM($\uparrow$) | LPIPS($\downarrow$) | FID($\downarrow$) |
| *Dirac*-PO (ours) | 26.67 | 0.7418 | **0.2716** | **53.36** | 25.41 | 0.7595 | 0.2611 | **39.43** |
| *Dirac*-DO (ours) | 28.47 | 0.8054 | 0.2972 | 69.15 | **26.98** | **0.8435** | **0.2234** | 51.87 |
| DPS Chung et al. (2022a) | 25.56 | 0.6878 | 0.3008 | 65.68 | 21.06 | 0.7238 | 0.2899 | 57.92 |
| DDRM Kawar et al. (2022a) | 27.21 | 0.7671 | 0.2849 | 65.84 | 25.62 | 0.8132 | 0.2313 | 54.37 |
| SwinIR Liang et al. (2021) | **28.53** | **0.8070** | 0.3048 | 72.93 | 24.46 | 0.8134 | 0.2660 | 59.94 |
| PnP-ADMM Chan et al. (2016) | 27.02 | 0.7596 | 0.3973 | 74.17 | 12.27 | 0.6205 | 0.4471 | 192.36 |
| ADMM-TV | 26.03 | 0.7323 | 0.4126 | 89.93 | 11.73 | 0.5618 | 0.5042 | 264.62 |

| Method | Deblurring | | | | Inpainting | | | |
|---|---|---|---|---|---|---|---|---|
| | PSNR($\uparrow$) | SSIM($\uparrow$) | LPIPS($\downarrow$) | FID($\downarrow$) | PSNR($\uparrow$) | SSIM($\uparrow$) | LPIPS($\downarrow$) | FID($\downarrow$) |
| *Dirac*-PO (ours) | 24.68 | 0.6582 | **0.3302** | 53.91 | 26.36 | 0.8087 | 0.2079 | 34.33 |
| *Dirac*-DO (ours) | **25.76** | **0.7085** | 0.3705 | 83.23 | **28.92** | **0.8958** | **0.1676** | 38.25 |
| DPS Chung et al. (2022a) | 21.51 | 0.5163 | 0.4235 | **52.60** | 22.71 | 0.8026 | 0.1986 | 34.55 |
| DDRM Kawar et al. (2022a) | 24.53 | 0.6676 | 0.3917 | 61.06 | 25.92 | 0.8347 | 0.2138 | **33.71** |
| PnP-ADMM Chan et al. (2016) | 25.02 | 0.6722 | 0.4565 | 98.72 | 18.14 | 0.7901 | 0.2709 | 101.25 |
| ADMM-TV | 24.31 | 0.6441 | 0.4578 | 88.26 | 17.60 | 0.7229 | 0.3157 | 120.22 |

Table 1: Experimental results on the FFHQ (top) and ImageNet (bottom) test splits.

et al. (2020) architecture. As the pre-trained score-models for competing methods have been trained on the full CelebA-HQ dataset, we test all methods for fair comparison on the first $1k$ images of the FFHQ Karras et al. (2019) dataset. For ImageNet experiments, we sample 1 image from each class from the official validation split to create disjoint validation and test sets of $1k$ images each. We only train our model on the train split of ImageNet.

We investigate two degradation processes of very different properties: Gaussian blur and inpainting. In all cases, Gaussian noise with $\sigma_1 = 0.05$ is added to the measurements in the $[0, 1]$ range. We use standard geometric noise scheduling with $\sigma_{max} = 0.05$ and $\sigma_{min} = 0.01$ in the SDP. For Gaussian blur, we use a kernel size of 61, with standard deviation of $w_{max} = 3$. We vary the standard deviation of the kernel between $w_{max}$(strongest) and $0.3$ (weakest) to parameterize the severity of Gaussian blur in the degradation process, and use the scheduling method described in Appendix D to specify $\mathcal{A}_t$. For inpainting, we generate a smooth mask in the form $\left(1 - \frac{f(\boldsymbol{x};w_t)}{\max_{\boldsymbol{x}} f(\boldsymbol{x};w_t)}\right)^k$, where $f(\boldsymbol{x};w_t)$ denotes the density of a zero-mean isotropic Gaussian with standard deviation $w_t$ that controls the size of the mask and $k = 4$ for sharper transition. We set $w_1 = 50$ for CelebA-HQ/FFHQ inpainting and 30 for ImageNet inpainting.

We compare our method against DDRM Kawar et al. (2022a), a well-established diffusion-based linear inverse problem solver; DPS Chung et al. (2022a), a recent, state-of-the-art diffusion technique for noisy inverse problems; SwinIR Liang et al. (2021), a state-of-the-art transformer-based supervised image restoration model; PnP-ADMM Chan et al. (2016), a reliable traditional solver with learned denoiser; and ADMM-TV, a classical optimization technique. To evaluate performance, we use PSNR and SSIM as distortion metrics and LPIPS and FID as perceptual quality metrics.

**Deblurring –** We train our model on $\mathcal{L}_{IR}(\Delta t = 0, \boldsymbol{\theta})$, as we observed no significant difference in using other incremental reconstruction losses, due to the smoothness of the degradation. We show results on our perception-optimized (PO) reconstructions, tuned for best LPIPS and our distortion-optimized (DO) reconstructions, tuned for best PSNR on a separate validation set via early-stopping

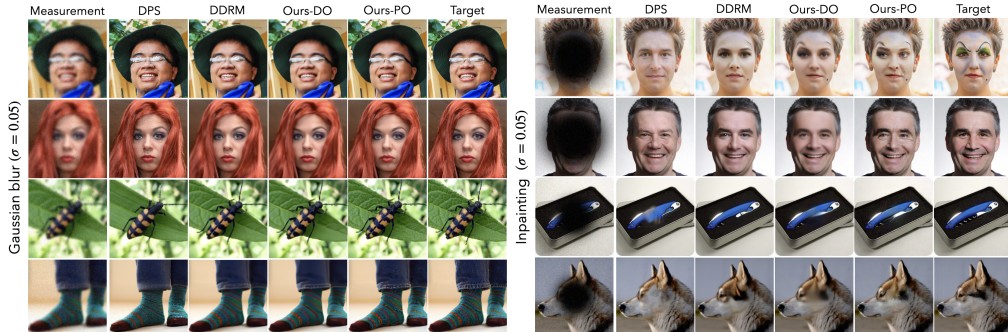

Figure 5: Visual comparison of reconstructions: Gaussian blur (left) and inpainting (right).

at the PSNR-peak (see Fig. 3). Our results, summarized in Table 1 (left side), demonstrate superior performance compared with other diffusion methods in terms of both distortion and perceptual metrics. Our DO model closely matches the distortion quality of SwinIR, a strong non-diffusion baseline known to outperform other diffusion solvers in terms of distortion metrics Chung et al. (2022a). Visual comparison in Figure 5 (left) reveals that DDRM produces reliable reconstructions, similar to our DO images, but they often lack detail. In contrast, DPS produces detailed images, similar to our PO reconstructions, but often with hallucinated details inconsistent with the measurement.

**Inpainting –** We train our model on $\mathcal{L}_{IR}(\Delta t = 1, \boldsymbol{\theta})$, as we see improvement in reconstruction quality as $\Delta t$ is increased. We hypothesize that this is due to sharp changes in the inpainting operator with respect to $t$, which can be mitigated by the incremental reconstruction loss according to Theorem 3.4. Ablations on the effect of $\Delta t$ in the incremental reconstruction loss can be found in Appendix I. We tuned models to optimize FID, as it is more suitable than pairwise image metrics to evaluate generated image content. Our results in Table 1 (right side) shows best performance in most metrics, followed by DDRM. Fig. 5 (right) shows, that our method generates high quality images even when limited context is available.

**Data consistency –** Consistency between reconstructions and the measurement is crucial in inverse problem solving. Our proposed method has the additional benefit of maintaining data consistency throughout the reverse process, as shown in Theorem 3.6 in the ideal case, however we empirically validate this claim. Figure 4 (left) shows the evolution of $\epsilon_{dc} := \|\tilde{\boldsymbol{y}} - \mathcal{A}_1(\hat{\boldsymbol{x}}_0(\boldsymbol{y}_t))\|^2$, where $\hat{\boldsymbol{x}}_0(\boldsymbol{y}_t)$ is the clean image estimate at time $t$ ($\Phi_{\boldsymbol{\theta}}(\boldsymbol{y}_t, t)$ for our method). Since $\tilde{\boldsymbol{y}} = \mathcal{A}_1(\boldsymbol{x}_0) + \sigma_1^2$, we expect $\epsilon_{dc}$ to approach $\sigma_1^2$ in case of perfect data consistency. We observe that our method, without applying guidance, stays close to the noise floor throughout the reverse process, while other techniques approach data consistency only close to $t = 1$. In case of DPS, we observe that data consistency is not always satisfied (see Figure 4, center), as DPS only guides the iterates towards data consistency, but does not directly enforce it. As our technique reverses an SDP, our intermediate reconstructions are always interpretable as degradations of varying severity of the same underlying image. This property allows us to early-stop the reconstruction and still obtain consistent reconstructions.

**Sampling speed –** *Dirac* requires low number of reverse diffusion steps for high quality reconstructions leading to fast sampling. Figure 4 (right) compares the perceptual quality at different number of reverse diffusion steps for diffusion-based inverse problem solvers. Our method typically requires $20 - 100$ steps for optimal perceptual quality, and shows the most favorable scaling in the low-NFE regime. Due to early-stopping we can trade-off perceptual quality for better distortion metrics and even further sampling speed-up. We obtain acceptable results even with one-shot reconstruction.

## 5 CONCLUSIONS AND LIMITATIONS

We propose a novel framework for solving inverse problems by reversing a stochastic degradation process. Our solver can flexibly trade off perceptual image quality for more traditional distortion metrics and sampling speedup. Moreover, we show both theoretically and empirically that our method maintains consistency with the measurement throughout the reverse process. *Dirac* produces reconstructions of exceptional quality in terms of both perceptual and distortion-based metrics, surpassing comparable state-of-the-art methods on multiple high-resolution datasets and image restoration tasks. The main limitation of our method is that a model needs to be trained from scratch for each inverse problem, whereas other diffusion-based solvers leverage pretrained score networks. Incorporating pretrained models into our framework is an interesting direction for future work.

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

APPENDIX

# A  PROOFS

## A.1  DENOISING SCORE-MATCHING GUARANTEE

Just as in standard diffusion, we approximate the score of the noisy, degraded data distribution $\nabla_{\boldsymbol{y}_t} q_t(\boldsymbol{y}_t)$ by matching the score of the tractable conditional distribution $\nabla_{\boldsymbol{y}_t} q_t(\boldsymbol{y}_t|\boldsymbol{x}_0)$ via minimizing the loss in (10). For standard Score-Based Models with $\mathcal{A}_t = \mathbf{I}$, the seminal work of Vincent (2011) guarantees that the true score is learned by denoising score-matching. More recently, Daras et al. (2022) points out that this result holds for a wide range of corruption processes, with the technical condition that the SDP assigns non-zero probability to all $\boldsymbol{y}_t$ for any given clean image $\boldsymbol{x}_0$. This condition is satisfied by adding Gaussian noise. For the sake of completeness, we include the theorem from Daras et al. (2022) updated with the notation from this paper.

**Theorem A.1.** *Let $q_0$ and $q_t$ be two distributions in $\mathbb{R}^n$. Assume that all conditional distributions, $q_t(\boldsymbol{y}_t|\boldsymbol{x}_0)$, are supported and differentiable in $\mathbb{R}^n$. Let:*

$$J_1(\theta) = \frac{1}{2}\mathbb{E}_{\boldsymbol{y}_t \sim q_t}\left[\|\boldsymbol{s}_\theta(\boldsymbol{y}_t, t) - \nabla_{\boldsymbol{y}_t}\log q_t(\boldsymbol{y}_t)\|^2\right], \tag{14}$$

$$J_2(\theta) = \frac{1}{2}\mathbb{E}_{(\boldsymbol{x}_0, \boldsymbol{y}_t) \sim q_0(\boldsymbol{x}_0)q_t(\boldsymbol{y}_t|\boldsymbol{x}_0)}\left[\|\boldsymbol{s}_\theta(\boldsymbol{y}_t, t) - \nabla_{\boldsymbol{y}_t}\log q_t(\boldsymbol{y}_t|\boldsymbol{x}_0)\|^2\right]. \tag{15}$$

*Then, there is a universal constant $C$ (that does not depend on $\theta$) such that: $J_1(\theta) = J_2(\theta) + C$.*

The proof, that follows the calculations of Vincent (2011), can be found in Appendix A.1. of Daras et al. (2022). This result implies that by minimizing the denoising score-matching objective in (15), the objective in (14) is also minimized, thus the true score is learned via matching the tractable conditional distribution $q_t(\boldsymbol{y}_t|\boldsymbol{x}_0)$ governing SDPs.

## A.2  THEOREM 3.4.

**Assumption A.2** (Lipschitzness of degradation). Assume that $\|\mathcal{A}_t(\boldsymbol{x}) - \mathcal{A}_t(\boldsymbol{y})\| \leq L_x^{(t)}\|\boldsymbol{x} - \boldsymbol{y}\|$, $\forall \boldsymbol{x}, \boldsymbol{y} \in \mathbb{R}^n$, $\forall t \in [0, 1]$ and $\|\mathcal{A}_{t'}(\boldsymbol{x}) - \mathcal{A}_{t''}(\boldsymbol{x})\| \leq L_t|t' - t''|$, $\forall \boldsymbol{x} \in \mathbb{R}^n$, $\forall t', t'' \in [0, 1]$.

**Assumption A.3** (Bounded signals). Assume that each entry of clean signals $\boldsymbol{x}_0$ are bounded as $\boldsymbol{x}_0[i] \leq B$, $\forall i \in (1, 2, ..., n)$.

**Lemma A.4.** *Assume $\boldsymbol{y}_t = \mathcal{A}_t(\boldsymbol{x}_0) + \boldsymbol{z}_t$ with $\boldsymbol{x}_0 \sim q_0(\boldsymbol{x}_0)$ and $\boldsymbol{z}_t \sim \mathcal{N}(0, \sigma_t^2\mathbf{I})$ and that Assumption A.2 holds. Then, the Jensen gap is upper bounded as $\|\mathbb{E}[\mathcal{A}_{t'}(\boldsymbol{x}_0)|\boldsymbol{y}_t] - \mathcal{A}_{t'}(\mathbb{E}[\boldsymbol{x}_0|\boldsymbol{y}_t])\| \leq L_x^{(t')}\sqrt{n}B$, $\forall t, t' \in [0, 1]$.*

*Proof.*

$$\|\mathbb{E}[\mathcal{A}_{t'}(\boldsymbol{x}_0)|\boldsymbol{y}_t] - \mathcal{A}_{t'}(\mathbb{E}[\boldsymbol{x}_0|\boldsymbol{y}_t])\| \overset{(1)}{\leq} \int \|\mathcal{A}_{t'}(\boldsymbol{x}_0) - \mathcal{A}_{t'}(\mathbb{E}[\boldsymbol{x}_0|\boldsymbol{y}_t])\| p(\boldsymbol{x}_0|\boldsymbol{y}_t)d\boldsymbol{x}_0$$

$$\overset{(2)}{\leq} \sqrt{\int \|\mathcal{A}_{t'}(\boldsymbol{x}_0) - \mathcal{A}_{t'}(\mathbb{E}[\boldsymbol{x}_0|\boldsymbol{y}_t])\|^2 p(\boldsymbol{x}_0|\boldsymbol{y}_t)d\boldsymbol{x}_0}$$

$$\leq L_x^{(t')}\sqrt{\int \|\boldsymbol{x}_0 - \mathbb{E}[\boldsymbol{x}_0|\boldsymbol{y}_t]\|^2 p(\boldsymbol{x}_0|\boldsymbol{y}_t)d\boldsymbol{x}_0}$$

$$\overset{(3)}{\leq} L_x^{(t')}\sqrt{\int \|\boldsymbol{x}_0\|^2 p(\boldsymbol{x}_0|\boldsymbol{y}_t)d\boldsymbol{x}_0}$$

$$\leq L_x^{(t')}\sqrt{\int nB^2 p(\boldsymbol{x}_0|\boldsymbol{y}_t)d\boldsymbol{x}_0} = L_x^{(t')}\sqrt{n}B$$

Here (1) and (2) hold due to Jensen's inequality, and in (3) we use the fact that $\mathbb{E}[\boldsymbol{x}_0|\boldsymbol{y}_t]$ is the minimum mean-squared error (MMSE) estimator of $\boldsymbol{x}_0$, thus we can replace it with 0 to get an upper bound. $\qquad\square$

**Theorem. 3.4** *Let $\hat{\mathcal{R}}(t, \Delta t; \boldsymbol{y}_t) = \mathcal{A}_{t-\Delta t}(\Phi_{\boldsymbol{\theta}}(\boldsymbol{y}_t, t)) - \mathcal{A}_t(\Phi_{\boldsymbol{\theta}}(\boldsymbol{y}_t, t))$ denote our estimate of the incremental reconstruction, where $\Phi_{\boldsymbol{\theta}}(\boldsymbol{y}_t, t)$ is trained on the loss in (13). Let $\mathcal{R}^*(t, \Delta t; \boldsymbol{y}_t) = \mathbb{E}[\mathcal{R}(t, \Delta t; \boldsymbol{x}_0)|\boldsymbol{y}_t]$ denote the MMSE estimator of $\mathcal{R}(t, \Delta t; \boldsymbol{x}_0)$. If Assumptions A.3 and A.2 hold and the error in our score network is bounded by $\|s_{\boldsymbol{\theta}}(\boldsymbol{y}_t, t) - \nabla_{\boldsymbol{y}_t} \log q_t(\boldsymbol{y}_t)\| \le \frac{\epsilon_t}{\sigma_t^2}, \ \forall t \in [0, 1]$, then*

$$\|\hat{\mathcal{R}}(t, \Delta t; \boldsymbol{y}_t) - \mathcal{R}^*(t, \Delta t; \boldsymbol{y}_t)\| \le (L_x^{(t)} + L_x^{(t-\Delta t)})\sqrt{n}B + 2L_t\Delta t + 2\epsilon_t.$$

*Proof.* First, we note that due to Tweedie's formula,

$$\mathbb{E}[\mathcal{A}_t(\boldsymbol{x}_0)|\boldsymbol{y}_t] = \boldsymbol{y}_t + \sigma_t^2 \nabla_{\boldsymbol{y}_t} \log q_t(\boldsymbol{y}_t).$$

Since we parameterized our score model as

$$s_{\boldsymbol{\theta}}(\boldsymbol{y}_t, t) = \frac{\mathcal{A}_t(\Phi_{\boldsymbol{\theta}}(\boldsymbol{y}_t, t)) - \boldsymbol{y}_t}{\sigma_t^2},$$

the assumption that $\|s_{\boldsymbol{\theta}}(\boldsymbol{y}_t, t) - \nabla_{\boldsymbol{y}_t} \log q_t(\boldsymbol{y}_t)\| \le \frac{\epsilon_t}{\sigma_t^2}$, is equivalent to

$$\|\mathcal{A}_t(\Phi_{\boldsymbol{\theta}}(\boldsymbol{y}_t, t)) - \mathbb{E}[\mathcal{A}_t(\boldsymbol{x}_0)|\boldsymbol{y}_t]\| \le \epsilon_t. \tag{16}$$

By applying the triangle inequality repeatedly, and applying Lemma A.4 and (16)

$$\left\| \hat{\mathcal{R}}(t, \Delta t; \boldsymbol{y}_t) - \mathcal{R}^*(t, \Delta t; \boldsymbol{y}_t) \right\|$$
$$= \|(\mathcal{A}_{t-\Delta t}(\Phi_{\boldsymbol{\theta}}(\boldsymbol{y}_t, t)) - \mathcal{A}_t(\Phi_{\boldsymbol{\theta}}(\boldsymbol{y}_t, t))) - (\mathbb{E}[\mathcal{A}_{t-\Delta t}(\boldsymbol{x}_0)|\boldsymbol{y}_t] - \mathbb{E}[\mathcal{A}_t(\boldsymbol{x}_0)|\boldsymbol{y}_t])\|$$
$$\le \|\mathcal{A}_{t-\Delta t}(\Phi_{\boldsymbol{\theta}}(\boldsymbol{y}_t, t)) - \mathbb{E}[\mathcal{A}_{t-\Delta t}(\boldsymbol{x}_0)|\boldsymbol{y}_t]\| + \|\mathcal{A}_t(\Phi_{\boldsymbol{\theta}}(\boldsymbol{y}_t, t)) - \mathbb{E}[\mathcal{A}_t(\boldsymbol{x}_0)|\boldsymbol{y}_t]\|$$
$$\le \|\mathcal{A}_{t-\Delta t}(\Phi_{\boldsymbol{\theta}}(\boldsymbol{y}_t, t)) - \mathcal{A}_{t-\Delta t}(\mathbb{E}[\boldsymbol{x}_0|\boldsymbol{y}_t]) + \mathcal{A}_{t-\Delta t}(\mathbb{E}[\boldsymbol{x}_0|\boldsymbol{y}_t]) - \mathbb{E}[\mathcal{A}_{t-\Delta t}(\boldsymbol{x}_0)|\boldsymbol{y}_t]\| + \epsilon_t$$
$$\le \|\mathcal{A}_{t-\Delta t}(\Phi_{\boldsymbol{\theta}}(\boldsymbol{y}_t, t)) - \mathcal{A}_{t-\Delta t}(\mathbb{E}[\boldsymbol{x}_0|\boldsymbol{y}_t])\| + L_x^{(t-\Delta t)}\sqrt{n}B + \epsilon_t$$
$$\le \|\mathcal{A}_{t-\Delta t}(\Phi_{\boldsymbol{\theta}}(\boldsymbol{y}_t, t)) - \mathcal{A}_t(\Phi_{\boldsymbol{\theta}}(\boldsymbol{y}_t, t))\| + \|\mathcal{A}_t(\Phi_{\boldsymbol{\theta}}(\boldsymbol{y}_t, t)) - \mathcal{A}_t(\mathbb{E}[\boldsymbol{x}_0|\boldsymbol{y}_t])\|$$
$$\quad + \|\mathcal{A}_t(\mathbb{E}[\boldsymbol{x}_0|\boldsymbol{y}_t]) - \mathcal{A}_{t-\Delta t}(\mathbb{E}[\boldsymbol{x}_0|\boldsymbol{y}_t])\| + L_x^{(t-\Delta t)}\sqrt{n}B + \epsilon_t$$
$$\le \|\mathcal{A}_t(\Phi_{\boldsymbol{\theta}}(\boldsymbol{y}_t, t)) - \mathcal{A}_t(\mathbb{E}[\boldsymbol{x}_0|\boldsymbol{y}_t])\| + 2L_t\Delta t + L_x^{(t-\Delta t)}\sqrt{n}B + \epsilon_t$$
$$\le \|\mathcal{A}_t(\Phi_{\boldsymbol{\theta}}(\boldsymbol{y}_t, t)) - \mathbb{E}[\mathcal{A}_t(\boldsymbol{x}_0)|\boldsymbol{y}_t]\| + \|\mathbb{E}[\mathcal{A}_t(\boldsymbol{x}_0)|\boldsymbol{y}_t] - \mathcal{A}_t(\mathbb{E}[\boldsymbol{x}_0|\boldsymbol{y}_t])\|$$
$$\quad + 2L_t\Delta t + L_x^{(t-\Delta t)}\sqrt{n}B + \epsilon_t$$
$$\le 2L_t\Delta t + (L_x^{(t-\Delta t)} + L_x^{(t)})\sqrt{n}B + 2\epsilon_t.$$

$\square$

We note that the appearance of $L_t$ in the upper bound provides a possible explanation why masking diffusion models are significantly worse in image generation than models relying on blurring, as observed in Daras et al. (2022). Masking leads to sharp jumps in pixel values at the border of the inpainting mask, thus $L_t$ can be arbitrarily large. This can be compensated to a certain degree by choosing a very small $\Delta t$ (very large number of sampling steps), which has also been observed in Daras et al. (2022).

### A.3 INCREMENTAL RECONSTRUCTION LOSS GUARANTEE

**Assumption A.5.** The forward degradation transition function $\mathcal{G}_{t' \to t''}$ for any $t', t'' \in [0, 1]$, $t' < t''$ is Lipschitz continuous: $\|\mathcal{G}_{t' \to t''}(\boldsymbol{x}) - \mathcal{G}_{t' \to t''}(\boldsymbol{y})\| \le L_G(t', t'')\|\boldsymbol{x} - \boldsymbol{y}\|$, $\forall t', t'' \in [0, 1]$, $t' < t''$, $\forall \boldsymbol{x}, \boldsymbol{y} \in \mathbb{R}^n$.

This is a very natural assumption, as we don't expect the distance between two images after applying a degradation to grow arbitrarily large.

**Proposition A.6.** *If the model $\Phi_{\boldsymbol{\theta}}(\boldsymbol{y}_t, t)$ has large enough capacity, such that $\mathcal{L}_{IR}(\Delta t, \boldsymbol{\theta}) = 0$ is achieved, then $s_{\boldsymbol{\theta}}(\boldsymbol{y}_t, t) = \nabla_{\boldsymbol{y}_t} \log q_t(\boldsymbol{y}_t)$, $\forall t \in [0, 1]$. Otherwise, if Assumption A.5 holds, then we have*

$$\mathcal{L}(\boldsymbol{\theta}) \le \max_{t \in [0,1]} (L_G(\tau, t)) \mathcal{L}_{IR}(\Delta t, \boldsymbol{\theta}). \tag{17}$$

*Proof.* We denote $\tau = \max(0, t - \Delta t)$. First, if $\mathcal{L}_{IR}(\Delta t, \boldsymbol{\theta}) = 0$, then

$$\mathcal{A}_\tau(\Phi_{\boldsymbol{\theta}}(\boldsymbol{y}_t, t)) = \mathcal{A}_\tau(\boldsymbol{x}_0)$$

for all $(\boldsymbol{x}_0, \boldsymbol{y}_t)$ such that $q_t(\boldsymbol{x}_0, \boldsymbol{y}_t) > 0$. Applying the forward degradation transition function to both sides yields

$$\mathcal{G}_{\tau \to t}(\mathcal{A}_\tau(\Phi_{\boldsymbol{\theta}}(\boldsymbol{y}_t, t))) = \mathcal{G}_{\tau \to t}(\mathcal{A}_\tau(\boldsymbol{x}_0)),$$

which is equivalent to

$$\mathcal{A}_t(\Phi_{\boldsymbol{\theta}}(\boldsymbol{y}_t, t)) = \mathcal{A}_t(\boldsymbol{x}_0).$$

This in turn means that $\mathcal{L}(\boldsymbol{\theta}) = 0$ and thus due to Theorem A.1 the score is learned.

In the more general case,

$$\begin{aligned}
\mathcal{L}(\boldsymbol{\theta}) &= \mathbb{E}_{t, (\boldsymbol{x}_0, \boldsymbol{y}_t)} \left[ w_t \left\| \mathcal{A}_t(\Phi_{\boldsymbol{\theta}}(\boldsymbol{y}_t, t)) - \mathcal{A}_t(\boldsymbol{x}_0) \right\|^2 \right] \\
&= \mathbb{E}_{t, (\boldsymbol{x}_0, \boldsymbol{y}_t)} \left[ w_t \left\| \mathcal{G}_{\tau \to t}(\mathcal{A}_\tau(\Phi_{\boldsymbol{\theta}}(\boldsymbol{y}_t, t))) - \mathcal{G}_{\tau \to t}(\mathcal{A}_\tau(\boldsymbol{x}_0)) \right\|^2 \right] \\
&\leq \mathbb{E}_{t, (\boldsymbol{x}_0, \boldsymbol{y}_t)} \left[ w_t L_G(\tau, t) \left\| \mathcal{A}_\tau(\Phi_{\boldsymbol{\theta}}(\boldsymbol{y}_t, t)) - \mathcal{A}_\tau(\boldsymbol{x}_0) \right\|^2 \right] \\
&\leq \max_{t \in [0,1]} (L_G(\tau, t)) \mathbb{E}_{t, (\boldsymbol{x}_0, \boldsymbol{y}_t)} \left[ w_t \left\| \mathcal{A}_\tau(\Phi_{\boldsymbol{\theta}}(\boldsymbol{y}_t, t)) - \mathcal{A}_\tau(\boldsymbol{x}_0) \right\|^2 \right] \\
&= \max_{t \in [0,1]} (L_G(\tau, t)) \mathcal{L}_{IR}(\Delta t, \boldsymbol{\theta})
\end{aligned}$$

$\square$

This means that if the model has large enough capacity, minimizing the incremental reconstruction loss in (12) also implies minimizing (10), and thus the true score is learned (denoising is achieved). Otherwise, the incremental reconstruction loss is an upper bound on the loss in (10). Training a model on (12), the model learns not only to denoise, but also to perform small, incremental reconstructions of the degraded image such that $\mathcal{A}_{t-\Delta t}(\Phi_{\boldsymbol{\theta}}(\boldsymbol{y}_t, t)) \approx \mathcal{A}_{t-\Delta t}(\boldsymbol{x}_0)$. There is however a trade-off between incremental reconstruction performance and learning the score: as Proposition A.6 indicates, we are optimizing an upper bound to (10) and thus it is possible that the score estimation is less accurate. We expect our proposed incremental reconstruction loss to work best in scenarios where the degradation may change rapidly with respect to $t$ and hence a network trained to accurately estimate $\mathcal{A}_t(\boldsymbol{x}_0)$ from $\boldsymbol{y}_t$ may become inaccurate when predicting $\mathcal{A}_{t-\Delta t}(\boldsymbol{x}_0)$ from $\boldsymbol{y}_t$. This hypothesis is further supported by our experiments in Section 4. Finally, we mention that in the extreme case where we choose $\Delta t = 1$, we obtain a loss function purely in clean image domain.

## A.4 THEOREM 3.7

**Lemma A.7** (Transitivity of data consistency). *If* $\boldsymbol{y}_{t^+} \overset{d.c.}{\sim} \boldsymbol{y}_t$ *and* $\boldsymbol{y}_{t^{++}} \overset{d.c.}{\sim} \boldsymbol{y}_{t^+}$ *with* $t < t^+ < t^{++}$, *then* $\boldsymbol{y}_{t^{++}} \overset{d.c.}{\sim} \boldsymbol{y}_t$.

*Proof.* By the definition of data consistency $\boldsymbol{y}_{t^{++}} \overset{d.c.}{\sim} \boldsymbol{y}_{t^+} \Rightarrow \exists \boldsymbol{x}_0 : \mathcal{A}_{t^{++}}(\boldsymbol{x}_0) = \boldsymbol{y}_{t^{++}}$ and $\mathcal{A}_{t^+}(\boldsymbol{x}_0) = \boldsymbol{y}_{t^+}$. On the other hand, $\boldsymbol{y}_{t^+} \overset{d.c.}{\sim} \boldsymbol{y}_t \Rightarrow \exists \boldsymbol{x}_0' : \mathcal{A}_{t^+}(\boldsymbol{x}_0') = \boldsymbol{y}_{t^+}$ and $\mathcal{A}_t(\boldsymbol{x}_0') = \boldsymbol{y}_t$. Therefore,

$$\boldsymbol{y}_{t^{++}} = \mathcal{A}_{t^{++}}(\boldsymbol{x}_0) = \mathcal{G}_{t^+ \to t^{++}}(\mathcal{A}_{t^+}(\boldsymbol{x}_0)) = \mathcal{G}_{t^+ \to t^{++}}(\boldsymbol{y}_{t^+}) = \mathcal{G}_{t^+ \to t^{++}}(\mathcal{A}_{t^+}(\boldsymbol{x}_0')) = \mathcal{A}_{t^{++}}(\boldsymbol{x}_0').$$

By the definition of data consistency, this implies $\boldsymbol{y}_{t^{++}} \overset{d.c.}{\sim} \boldsymbol{y}_t$. $\square$

**Theorem. 3.7.** *Assume that we run the updates in* (6) *with* $s_{\boldsymbol{\theta}}(\boldsymbol{y}_t, t) = \nabla_{\boldsymbol{y}_t} \log q_t(\boldsymbol{y}_t | \boldsymbol{x}_0)$, $\forall t \in [0, 1]$ *and* $\hat{\mathcal{R}}(t, \Delta t; \boldsymbol{y}_t) = \mathcal{R}(t, \Delta t; \boldsymbol{x}_0)$, $\boldsymbol{x}_0 \in \mathcal{X}_0$. *If we start from a noisy degraded observation* $\tilde{\boldsymbol{y}} = \mathcal{A}_1(\boldsymbol{x}_0) + \boldsymbol{z}_1$, $\boldsymbol{x}_0 \in \mathcal{X}_0$, $\boldsymbol{z}_1 \sim \mathcal{N}(\boldsymbol{0}, \sigma_1^2 \boldsymbol{I})$ *and run the updates in* (6) *for* $\tau = 1, 1 - \Delta t, ..., \Delta t, 0$, *then we have*

$$\mathbb{E}[\tilde{\boldsymbol{y}}] \overset{d.c.}{\sim} \mathbb{E}[\boldsymbol{y}_\tau], \ \forall \tau \in [1, 1 - \Delta t, ..., \Delta t, 0]. \tag{18}$$

*Proof.* Assume that we start from a known measurement $\tilde{\boldsymbol{y}} := \boldsymbol{y}_t = \mathcal{A}_t(\boldsymbol{x}_0) + \boldsymbol{z}_t$ at arbitrary time $t$ and run reverse diffusion from $t$ with time step $\Delta t$. Starting from $t = 1$ that we have looked at in the paper is a subset of this problem. Starting from arbitrary $\boldsymbol{y}_t$, the first update takes the form

$$
\begin{aligned}
\boldsymbol{y}_{t-\Delta t} &= \boldsymbol{y}_t + \mathcal{A}_{t-\Delta t}(\Phi_{\boldsymbol{\theta}}(\boldsymbol{y}_t, t)) - \mathcal{A}_t(\Phi_{\boldsymbol{\theta}}(\boldsymbol{y}_t, t)) \\
&\quad - (\sigma_{t-\Delta t}^2 - \sigma_t^2)\frac{\mathcal{A}_t(\Phi_{\boldsymbol{\theta}}(\boldsymbol{y}_t, t)) - \boldsymbol{y}_t}{\sigma_t^2} + \sqrt{\sigma_t^2 - \sigma_{t-\Delta t}^2}\boldsymbol{z} \\
&= \mathcal{A}_t(\boldsymbol{x}_0) + \boldsymbol{z}_t + \mathcal{A}_{t-\Delta t}(\Phi_{\boldsymbol{\theta}}(\boldsymbol{y}_t, t)) - \mathcal{A}_t(\Phi_{\boldsymbol{\theta}}(\boldsymbol{y}_t, t)) \\
&\quad - (\sigma_{t-\Delta t}^2 - \sigma_t^2)\frac{\mathcal{A}_t(\Phi_{\boldsymbol{\theta}}(\boldsymbol{y}_t, t)) - \mathcal{A}_t(\boldsymbol{x}_0) - \boldsymbol{z}_t}{\sigma_t^2} + \sqrt{\sigma_t^2 - \sigma_{t-\Delta t}^2}\boldsymbol{z}
\end{aligned}
$$

Due to our assumption on learning the score function, we have $\mathcal{A}_t(\Phi_{\boldsymbol{\theta}}(\boldsymbol{y}_t, t)) = \mathcal{A}_t(\boldsymbol{x}_0)$ and due to the perfect incremental reconstruction assumption $\mathcal{A}_{t-\Delta t}(\Phi_{\boldsymbol{\theta}}(\boldsymbol{y}_t, t)) = \mathcal{A}_{t-\Delta t}(\boldsymbol{x}_0)$. Thus, we have

$$
\boldsymbol{y}_{t-\Delta t} = \mathcal{A}_{t-\Delta t}(\boldsymbol{x}_0) + \frac{\sigma_{t-\Delta t}^2}{\sigma_t^2}\boldsymbol{z}_t + \sqrt{\sigma_t^2 - \sigma_{t-\Delta t}^2}\boldsymbol{z}.
$$

Since $z$ and $\boldsymbol{z}_t$ are independent Gaussian, we can combine the noise terms to yield

$$
\boldsymbol{y}_{t-\Delta t} = \mathcal{A}_{t-\Delta t}(\boldsymbol{x}_0) + \boldsymbol{z}_{t-\Delta t}, \tag{19}
$$

with $\boldsymbol{z}_{t-\Delta_t} \sim \mathcal{N}(\mathbf{0}, \left[\left(\frac{\sigma_{t-\Delta t}^2}{\sigma_t}\right)^2 + \sigma_t^2 - \sigma_{t-\Delta t}^2\right]\mathbf{I})$. This form is identical to the expression on our original measurement $\tilde{\boldsymbol{y}} = \boldsymbol{y}_t = \mathcal{A}_t(\boldsymbol{x}_0) + \boldsymbol{z}_t$, but with slightly lower degradation severity and noise variance. It is also important to point out that $\mathbb{E}[\boldsymbol{y}_t] \overset{d.c.}{\sim} \mathbb{E}[\boldsymbol{y}_{t-\Delta t}]$. If we repeat the update to find $\boldsymbol{y}_{t-2\Delta t}$, we will have the same form as in (19) and $\mathbb{E}[\boldsymbol{y}_{t-\Delta t}] \overset{d.c.}{\sim} \mathbb{E}[\boldsymbol{y}_{t-2\Delta t}]$. Due to the transitive property of data consistency (Lemma A.7), we also have $\mathbb{E}[\boldsymbol{y}_t] \overset{d.c.}{\sim} \mathbb{E}[\boldsymbol{y}_{t-2\Delta t}]$, that is data consistency is preserved with the original measurement. This reasoning can be then extended for every further update using the transitivity property, therefore we have data consistency in each iteration. $\qquad\square$

## B GUIDANCE

So far, we have only used our noisy observation $\tilde{\boldsymbol{y}} = \mathcal{A}_1(\boldsymbol{x}_0) + \boldsymbol{z}_1$ as a starting point for the reverse diffusion process, however the measurement is not used directly in the update in (6). We learned the score of the prior distribution $\nabla_{\boldsymbol{y}_t} \log q_t(\boldsymbol{y}_t)$, which we can leverage to sample from the posterior distribution $q_t(\boldsymbol{y}_t|\tilde{\boldsymbol{y}})$. In fact, using Bayes rule the score of the posterior distribution can be written as

$$
\nabla_{\boldsymbol{y}_t} \log q_t(\boldsymbol{y}_t|\tilde{\boldsymbol{y}}) = \nabla_{\boldsymbol{y}_t} \log q_t(\boldsymbol{y}_t) + \nabla_{\boldsymbol{y}_t} \log q_t(\tilde{\boldsymbol{y}}|\boldsymbol{y}_t), \tag{20}
$$

where we already approximate $\nabla_{\boldsymbol{y}_t} \log q_t(\boldsymbol{y}_t)$ via $s_{\boldsymbol{\theta}}(\boldsymbol{y}_t, t)$. Finding the posterior distribution analytically is not possible, and therefore we use the approximation $q_t(\tilde{\boldsymbol{y}}|\boldsymbol{y}_t) \approx q_t(\tilde{\boldsymbol{y}}|\Phi_{\boldsymbol{\theta}}(\boldsymbol{y}_t, t))$, from which distribution we can easily sample from. Since $q_t(\tilde{\boldsymbol{y}}|\Phi_{\boldsymbol{\theta}}(\boldsymbol{y}_t, t)) \sim \mathcal{N}(\mathcal{A}_1(\Phi_{\boldsymbol{\theta}}(\boldsymbol{y}_t, t)), \sigma_1^2\mathbf{I})$, our estimate of the posterior score takes the form

$$
s_{\boldsymbol{\theta}}'(\boldsymbol{y}_t, t) = s_{\boldsymbol{\theta}}(\boldsymbol{y}_t, t) - \eta_t \nabla_{\boldsymbol{y}_t}\frac{\|\tilde{\boldsymbol{y}} - \mathcal{A}_1(\Phi_{\boldsymbol{\theta}}(\boldsymbol{y}_t, t))\|^2}{2\sigma_1^2}, \tag{21}
$$

where $\eta_t$ is a hyperparameter that tunes how much we rely on the original noisy measurement. Even though we do not need to rely on $\tilde{\boldsymbol{y}}$ after the initial update for our method to work, we observe small improvements by adding the above guidance scheme to our algorithm.

For the sake of simplicity, in this discussion we merge the scaling of the gradient into the step size parameter as follows:

$$
s_{\boldsymbol{\theta}}'(\boldsymbol{y}_t, t) = s_{\boldsymbol{\theta}}(\boldsymbol{y}_t, t) - \eta_t' \nabla_{\boldsymbol{y}_t}\|\tilde{\boldsymbol{y}} - \mathcal{A}_1(\Phi_{\boldsymbol{\theta}}(\boldsymbol{y}_t, t))\|^2 \tag{22}
$$

We experiment with two choices of step size scheduling for the guidance term $\eta_t'$:

- *Standard deviation scaled (constant)*: $\eta_t = \eta\frac{1}{2\sigma_1^2}$, where $\eta$ is a constant hyperparameter and $\sigma_1^2$ is the noise level on the measurements. This scaling is justified by our derivation of the posterior score approximation, and matches (22).

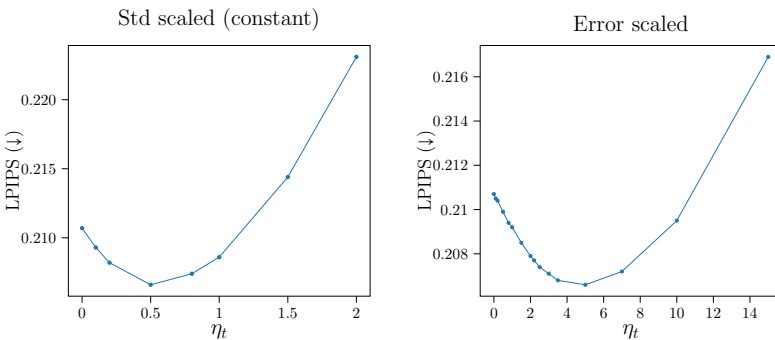

Figure 6: Effect of guidance step size on best reconstruction in terms of LPIPS. We perform experiments on the CelebA-HQ validation set on the deblurring task.

- *Error scaled*: $\eta_t = \eta \frac{1}{\|\tilde{\boldsymbol{y}} - \mathcal{A}_1(\Phi_{\boldsymbol{\theta}}(\boldsymbol{y}_t, t))\|}$, which has been proposed in Chung et al. (2022a). This method attempts to normalize the gradient of the data consistency term.

In general, we find that constant step size works better for deblurring, whereas error scaling performed slightly better for inpainting experiments, however the difference is minor. Figure 6 shows the results of our ablation study on the effect of $\eta_t$. We perform deblurring experiments on the CelebA-HQ validation set and plot the mean LPIPS (lower the better) with different step size scheduling methods and varying step size. We see some improvement in LPIPS when adding guidance to our method, however it is not a crucial component in obtaining high quality reconstructions, or for maintaining data-consistency.

## C  OVERVIEW OF THE ALGORITHM

A complete algorithmic overview of *Dirac* is depicted in Algorithm 1.

---

**Algorithm 1** *Dirac*

---

**Input:** $\tilde{\boldsymbol{y}}$: noisy observation, $\Phi_{\boldsymbol{\theta}}$: score network, $\mathcal{A}_t(\cdot)$: degradation function, $\Delta t$: step size, $\sigma_t$: noise std at time $t$, $\eta_t$: guidance step size, $\forall t \in [0, 1]$, $t_{stop}$: early-stopping parameter
    $N \leftarrow \lfloor 1/\Delta t \rfloor$
    $\boldsymbol{y} \leftarrow \tilde{\boldsymbol{y}}$
    **for** $i = 1$ to $N$ **do**
        $t \leftarrow 1 - \Delta t \cdot i$
        **if** $t \leq t_{stop}$ **then**                             ▷ Early-stopping
            **break**
        **end if**
        $\boldsymbol{z} \sim \mathcal{N}(0, \sigma_t^2 \mathbf{I})$
        $\hat{\boldsymbol{x}}_0 \leftarrow \Phi_{\boldsymbol{\theta}}(\boldsymbol{y}, t)$                      ▷ Predict posterior mean
        $\boldsymbol{y}_r \leftarrow \mathcal{A}_{t-\Delta t}(\hat{\boldsymbol{x}}_0) - \mathcal{A}_t(\hat{\boldsymbol{x}}_0)$       ▷ Incremental reconstruction
        $\boldsymbol{y}_d \leftarrow -\frac{\sigma_{t-\Delta t}^2 - \sigma_t^2}{\sigma_t^2}(\mathcal{A}_t(\hat{\boldsymbol{x}}_0) - \boldsymbol{y})$      ▷ Denoising
        $\boldsymbol{y}_g \leftarrow (\sigma_{t-\Delta t}^2 - \sigma_t^2)\nabla_y\|\tilde{\boldsymbol{y}} - \mathcal{A}_1(\hat{\boldsymbol{x}}_0)\|^2$      ▷ Guidance
        $\boldsymbol{y} \leftarrow \boldsymbol{y} + \boldsymbol{y}_r + \boldsymbol{y}_d + \eta_t \boldsymbol{y}_g + \sqrt{\sigma_t^2 - \sigma_{t-\Delta t}^2}\boldsymbol{z}$
    **end for**
**Output:** $\boldsymbol{y}$                            ▷ Alternatively, output $\hat{\boldsymbol{x}}_0$

---

## D  DEGRADATION SCHEDULING

When solving inverse problems, we have access to a noisy measurement $\tilde{\boldsymbol{y}} = \mathcal{A}(\boldsymbol{x}_0) + \boldsymbol{z}$ and we would like to find the corresponding clean image $\boldsymbol{x}_0$. In order to deploy our method, we need to

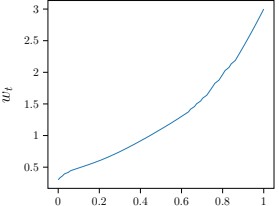 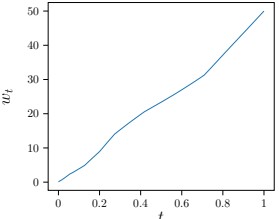 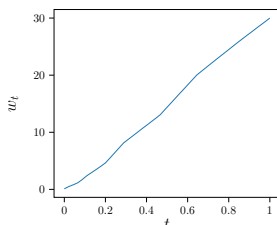

Figure 7: Results of degradation scheduling from Algorithm 2. Left: Gaussian blur with kernel std $w_t$ on CelebA-HQ. Center: inpainting with Gaussian mask with kernel width $w_t$ on CelebA-HQ. Right: inpainting with Gaussian mask on ImageNet.

define how the degradation changes with respect to severity $t$ following the properties specified in Definition 3.3. That is, we have to determine how to interpolate between the identity mapping $\mathcal{A}_0(\boldsymbol{x}) = \boldsymbol{x}$ for $t = 0$ and the most severe degradation $\mathcal{A}_1(\cdot) = \mathcal{A}(\cdot)$ for $t = 1$. Theorem 3.4 suggests that sharp changes in the degradation function with respect to $t$ should be avoided, however a more principled method of scheduling is needed.

In the context of image generation, Daras et al. (2022) proposes a scheduling framework that splits the path between the distribution of clean images $\mathcal{D}_0$ and the distribution of pure noise $\mathcal{D}_1$ into $T$ candidate distributions $\mathcal{D}_i$, $i \in [1/T, 2/T, ..., \frac{T-1}{T}]$. Then, they find a path through the candidate distributions that minimizes the total path length, where the distance between $\mathcal{D}_i$ and $\mathcal{D}_j$ is measured by the Wasserstein-distance. However, for image reconstruction, instead of distance between image distributions, we are more interested in how much a given image degrades in terms of image quality metrics such as PSNR or LPIPS. Therefore, we replace the Wasserstein-distance by a notion of distance between two degradation severities $d(t_i, t_j) := \mathbb{E}_{\boldsymbol{x}_0 \sim \mathcal{D}_0}[\mathcal{M}(\mathcal{A}_{t_i}(\boldsymbol{x}_0), \mathcal{A}_{t_j}(\boldsymbol{x}_0))]$, where $\mathcal{M}$ is some distortion-based or perceptual image quality metric that acts on a corresponding pair of images.

We propose a greedy algorithm to select a set of degradations from the set of candidates based on the above notion of dataset-dependent distance, such that the maximum distance is minimized. That is, our scheduler is not only a function of the degradation $\mathcal{A}_t$, but also the data. The intuitive reasoning to minimize the maximum distance is that our model has to be imbued with enough capacity to bridge the gap between any two consecutive distributions during the reverse process, and thus the most challenging transition dictates the required network capacity. In particular, given a budget of $m$ intermediate distributions on $[0, 1]$, we would like to pick a set of $m$ interpolating severities $\mathcal{S}$ such that

$$\mathcal{S} = arg \min_{\mathcal{T}} \max_i d(t_i, t_{i+1}), \tag{23}$$

where $\mathcal{T} = \{t_1, t_2, ..., t_m | t_i \in [0, 1], \ t_i < t_{i+1} \ \forall i \in (1, 2, ..., m)\}$ is the set of possible interpolating severities with budget $m$. To this end, we start with $\mathcal{S} = \{0, 1\}$ and add new interpolating severities one-by-one, such that the new point splits the interval in $\mathcal{S}$ with the maximum distance. Thus, over iterations the maximum distance is non-increasing. We also have local optimality, as moving a single interpolating severity must increase the maximum distance by the construction of the algorithm. Finally, we use linear interpolation in between the selected interpolating severities. The technique is summarized in Algorithm 2, and we refer the reader to the source code for implementation details.

The results of our proposed greedy scheduling algorithm are shown in Figure 7, where the distance is defined based on the LPIPS metric. In case of blurring, we see a sharp decrease in degradation severity close to $t = 1$. This indicates, that LPIPS difference between heavily blurred images is small, therefore most of the diffusion takes place at lower blur levels. On the other hand, we find that inpainting mask size is scaled almost linearly by our algorithm on both datasets we investigated.

## E    NOTE ON THE OUTPUT OF THE ALGORITHM

In the ideal case, $\sigma_0 = 0$ and $\mathcal{A}_0 = \mathbf{I}$. However, in practice due to geometric noise scheduling (e.g. $\sigma_0 = 0.01$), there is small magnitude additive noise expected on the final iterate. Moreover, in order to keep the scheduling of the degradation smooth, and due to numerical stability in practice

---

**Algorithm 2** Greedy Degradation Scheduling

---

**Input:** $\mathcal{M}$: pairwise image dissimilarity metric, $\mathcal{X}_0$: clean samples, $\mathcal{A}_t$: unscheduled degradation function, $N$: number of candidate points, $m$: number of interpolation points

$ts \leftarrow (0, \frac{1}{N-1}, \frac{2}{N-1}, ..., \frac{N-2}{N-1}, 1)$      $\triangleright$ $N$ candidate severities uniformly distributed over $[0, 1]$

$\mathcal{S} \leftarrow (1, N)$      $\triangleright$ Array of indices of output severities in $ts$

$d_{max} \leftarrow Distance(ts[1], ts[N])$ $\triangleright$ Maximum distance between two severities in the output array

$e_{start} \leftarrow 1$      $\triangleright$ Start index of edge with maximum distance

$e_{end} \leftarrow N$      $\triangleright$ End index of edge with maximum distance

**for** $i = 1$ to $m$ **do**

    $s \leftarrow FindBestSplit(e_{start}, e_{end}, d_{max})$

    $Append(\mathcal{S}, s)$

    $d_{max}, e_{start}, e_{end} \leftarrow UpdateMax(\mathcal{S})$

**end for**

**Output:** $\mathcal{S}$

 

**procedure** DISTANCE$(t_i, t_j)$      $\triangleright$ Distance between degradation severities $t_i$ and $t_j$

    $d \leftarrow \frac{1}{|\mathcal{X}_0|} \sum_{x \in \mathcal{X}_0} \mathcal{M}(\mathcal{A}_{t_i}(x), \mathcal{A}_{t_j}(x))$

**Output:** $d$

**end procedure**

 

**procedure** FINDBESTSPLIT$(e_{start}, e_{end}, d_{max})$    $\triangleright$ Split edge into two new edges with minimal maximum distance

    $MaxDistance \leftarrow d_{max}$

    **for** $j = e_{start} + 1$ to $e_{end} - 1$ **do**

        $d_1 \leftarrow Distance(ts[e_{start}], ts[j])$

        $d_2 \leftarrow Distance(ts[j], ts[e_{end}])$

        **if** $\max(d_1, d_2) < MaxDistance$ **then**

            $MaxDistance \leftarrow \max(d_1, d_2)$

            $Split \leftarrow j$

        **end if**

    **end for**

**Output:** $Split$

**end procedure**

 

**procedure** UPDATEMAX$(\mathcal{S})$

    $MaxDistance \leftarrow 0$

    **for** $i = 1$ to $|S| - 1$ **do**

        $e_{start} \leftarrow \mathcal{S}[i]$

        $e_{end} \leftarrow \mathcal{S}[i+1]$

        $d \leftarrow Distance(ts[e_{start}], ts[e_{end}])$

        **if** $d > MaxDistance$ **then**

            $MaxDistance \leftarrow d$

            $NewStart \leftarrow e_{start}$

            $NewEnd \leftarrow e_{end}$

        **end if**

    **end for**

**Output:** $MaxDistance, NewStart, NewEnd$

**end procedure**

---

$\mathcal{A}_0$ may slightly deviate from the identity mapping close to $t = 0$ (for example very small amount of blur). Thus, even close to $t = 0$, there may be a gap between the iterates $\boldsymbol{y}_t$ and the posterior mean estimates $\hat{\boldsymbol{x}}_0 = \Phi_\theta(\boldsymbol{y}_t, t)$. Due to these reasons, we observe that in some experiments taking $\Phi_\theta(\boldsymbol{y}_t, t)$ as the final output yields better reconstructions. In case of early stopping, taking $\hat{\boldsymbol{x}}_0$ as the output is instrumental, as an intermediate iterate $\boldsymbol{y}_t$ represents a sample from the reverse SDP, thus it is expected to be noisy and degraded. However, as $\Phi_\theta(\boldsymbol{y}_t, t)$ always predicts the clean image, it can be used at any time step $t$ to obtain an early-stopped prediction of $\boldsymbol{x}_0$.

## F  EXPERIMENTAL DETAILS

**Datasets –** We evaluate our method on CelebA-HQ ($256 \times 256$) Karras et al. (2018) and ImageNet ($256 \times 256$) Deng et al. (2009). For CelebA-HQ training, we use $80\%$ of the dataset for training, and the rest for validation and testing. For ImageNet experiments, we sample $1$ image from each class from the official validation split to create disjoint validation and test sets of $1k$ images each. We only train our model on the official train split of ImageNet. We center-crop and resize ImageNet images to $256 \times 256$ resolution. For both datasets, we scale images to $[0, 1]$ range.

**Comparison methods –** We compare our method against DDRM Kawar et al. (2022a), the most well-established diffusion-based linear inverse problem solver; DPS Chung et al. (2022a), a very recent, state-of-the-art diffusion technique for noisy and possibly nonlinear inverse problems; PnP-ADMM Chan et al. (2016), a reliable traditional solver with learned denoiser; and ADMM-TV, a classical optimization technique. Furthermore, we perform comparison with InDI Delbracio & Milanfar (2023) in Section G. More details on comparison methods can be found in Section J.1.

**Models –** For *Dirac*, we train new models from scratch using the NCSN++Song et al. (2020) architecture with `67M` parameters for all tasks except for ImageNet inpainting, for which we scale the model to `126M` parameters. For competing methods that require a score model, we use pre-trained SDE-VP models[1] (`126M` parameters for CelebA-HQ, `553M` parameters for ImageNet). The architectural hyper-parameters for the various score-models can be seen in Table 2.

**Training details –** We train all models with Adam optimizer, with learning rate $0.0001$ and batch size 32 on $8\times$ Titan RTX GPUs, with the exception of the large model used for ImageNet inpainting experiments which we trained on $8\times$ A6000 GPUs. We only use exponential moving averaging for this large model. We train for approximately $10M$ examples seen by the network. For the weighting factor $w(t)$ in the loss, we set $w(t) = \frac{1}{\sigma_t^2}$ in all experiments.

**Degradations –** We investigate two degradation processes of very different properties: Gaussian blur and inpainting, both with additive Gaussian noise. In all cases, noise with $\sigma_1 = 0.05$ is added to the measurements in the $[0, 1]$ range. We use standard geometric noise scheduling with $\sigma_{max} = 0.05$ and $\sigma_{min} = 0.01$ in the SDP. For Gaussian blur, we use a kernel size of $61$, with standard deviation of $w_{max} = 3$ to create the measurements. We change the standard deviation of the kernel between $w_{max}$ (strongest) and $w_{min} = 0.3$ (weakest) to parameterize the severity of Gaussian blur in the degradation process, and use the scheduling method described in Section D to specify $\mathcal{A}_t$. In particular, we set

$$\mathcal{A}_t^{blur}(\boldsymbol{x}) = \boldsymbol{C}^{\Psi_t}\boldsymbol{x},$$

where $\boldsymbol{C}^{\Psi_t}$ is a matrix representing convolution with the Gaussian kernel $\Psi_t$. The degradation level is parameterized by the standard deviation of $\Psi_t$, and scheduled between $w_{max} = 3.0$ at $t = 1$ and $w_{min} = 0.3$ at $t = 0$. We keep an imperceptible amount of blur for $t = 0$ to avoid numerical instability with very small kernel widths. For inpainting, we generate a smooth mask in the form $\boldsymbol{M_t} = \left(1 - \frac{f(\boldsymbol{x};w_t)}{\max_{\boldsymbol{x}} f(\boldsymbol{x};w_t)}\right)^k$, where $f(\boldsymbol{x}; w_t)$ denotes the density of a zero-mean isotropic Gaussian with standard deviation $w_t$ that controls the size of the mask and $k = 4$ for sharper transition. That is, the degradation process is defined as

$$\mathcal{A}_t^{inpaint}(\boldsymbol{x}) = \boldsymbol{M}_t\boldsymbol{x}.$$

We set $w_1 = 50$ for CelebA-HQ/FFHQ inpainting and 30 for ImageNet inpainting, and set $\boldsymbol{M}_0 = \boldsymbol{I}$ in all experiments. We determine the schedule of $w_t$ for $t \in (0, 1)$ using Algorithm 2.

---

[1]CelebA-HQ: `https://github.com/ermongroup/SDEdit`
ImageNet: `https://github.com/openai/guided-diffusion`

| | *Dirac*(Ours) | | | |
|---|---|---|---|---|
| **Hparam** | Deblur/CelebA-HQ | Deblur/ImageNet | Inpainting/CelebA-HQ | Inpainting/ImageNet |
| model_channels | 128 | 128 | 128 | 256 |
| channel_mult | $[1, 1, 2, 2, 2, 2, 2]$ | $[1, 1, 2, 2, 2, 2, 2]$ | $[1, 1, 2, 2, 2, 2, 2]$ | $[1, 1, 2, 2, 4, 4]$ |
| num_res_blocks | 2 | 2 | 2 | 2 |
| attn_resolutions | $[16]$ | $[16]$ | $[16]$ | $[16]$ |
| dropout | 0.1 | 0.1 | 0.1 | 0.0 |
| Total # of parameters | 67M | 67M | 67M | 520M |

| | **DDRM/DPS** | | | |
|---|---|---|---|---|
| **Hparam** | Deblur/CelebA-HQ | Deblur/ImageNet | Inpainting/CelebA-HQ | Inpainting/ImageNet |
| model_channels | 128 | 256 | 128 | 256 |
| channel_mult | $[1, 1, 2, 2, 4, 4]$ | $[1, 1, 2, 2, 4, 4]$ | $[1, 1, 2, 2, 4, 4]$ | $[1, 1, 2, 2, 4, 4]$ |
| num_res_blocks | 2 | 2 | 2 | 2 |
| attn_resolutions | $[16]$ | $[32, 16, 8]$ | $[16]$ | $[32, 16, 8]$ |
| dropout | 0.0 | 0.0 | 0.0 | 0.0 |
| Total # of parameters | 126M | 553M | 126M | 553M |

Table 2: Architectural hyper-parameters for the score-models for *Dirac* (top) and other diffusion-based methods (bottom) in our experiments.

| | PO Sampling hyper-parameters | | | |
|---|---|---|---|---|
| **Hparam** | Deblur/CelebA-HQ | Deblur/ImageNet | Inpainting/CelebA-HQ | Inpainting/ImageNet |
| $\Delta t$ | 0.02 | 0.02 | 0.005 | 0.01 |
| $t_{stop}$ | 0.25 | 0.0 | 0.0 | 0.0 |
| $\eta_t$ | 0.5 | 0.2 | 1.0 | 0.0 |
| Guidance scaling | std | std | error | - |
| Output | $\hat{\boldsymbol{x}}_0$ | $\hat{\boldsymbol{x}}_0$ | $\boldsymbol{y}_t$ | $\boldsymbol{y}_t$ |

| | DO Sampling hyper-parameters | | | |
|---|---|---|---|---|
| **Hparam** | Deblur/CelebA-HQ | Deblur/ImageNet | Inpainting/CelebA-HQ | Inpainting/ImageNet |
| $\Delta t$ | 0.02 | 0.02 | 0.005 | 0.01 |
| $t_{stop}$ | 0.98 | 0.7 | 0.995 | 0.99 |
| $\eta_t$ | 0.5 | 1.5 | 1.0 | 0.0 |
| Guidance scaling | std | std | error | - |
| Output | $\hat{\boldsymbol{x}}_0$ | $\hat{\boldsymbol{x}}_0$ | $\hat{\boldsymbol{x}}_0$ | $\hat{\boldsymbol{x}}_0$ |

Table 3: Settings for perception optimized (PO) and distortion optimized (DO) sampling for all experiments on test data.

**Evaluation method** – To evaluate performance, we use PSNR and SSIM as distortion metrics and LPIPS and FID as perceptual quality metrics. For the final reported results, we scale and clip all outputs to the $[0, 1]$ range before computing the metrics. We use validation splits to tune the hyper-parameters for all methods, where we optimize for best LPIPS in the deblurring task and for best FID for inpainting. As the pre-trained score-models for competing methods have been trained on the full CelebA-HQ dataset, we test all methods for fair comparison on the first $1k$ images of the FFHQ Karras et al. (2019) dataset. The list of test images for ImageNet can be found in the source code.

**Sampling hyperparameters** – The settings are summarized in Table 3. We tune the reverse process hyper-parameters on validation data. For the interpretation of 'guidance scaling' we refer the reader to the explanation of guidance step size methods in Section B. In Table 3, 'output' refers to whether the final reconstruction is the last model output (posterior mean estimate, $\hat{\boldsymbol{x}}_0 = \Phi_\theta(\boldsymbol{y}_t, t)$) or the final iterate $\boldsymbol{y}_t$.

## G    COMPARISON WITH BLENDING

Our proposed method interpolates between degraded and clean distributions via a SDP. A parallel line of work Delbracio & Milanfar (2023); Heitz et al. (2023) considers an alternative formulation in which the intermediate distributions are convex combinations of degraded-clean image pairs, that is $\boldsymbol{y_t} = t\tilde{\boldsymbol{y}} + (1 - t)\boldsymbol{x_0}$. We compare the InDI Delbracio & Milanfar (2023) formulation to *Dirac* on the FFHQ dataset (Table 4). We observe comparable results on the deblurring task, however the

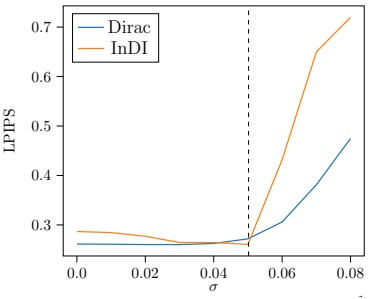

| Method | Deblurring | | Inpainting | |
|---|---|---|---|---|
| | LPIPS($\downarrow$) | FID($\downarrow$) | LPIPS($\downarrow$) | FID($\downarrow$) |
| Blending (InDI Delbracio & Milanfar (2023)) | **0.2604** | 56.27 | **0.2424** | 54.08 |
| Dirac-PO (ours) | 0.2716 | **53.36** | 0.2626 | **39.43** |

Table 4: Comparison with blending schedule on the FFHQ test split.

Figure 8: Robustness experiment: we simulate a mismatch between train and test noise levels (FFHQ test split, deblurring). Dirac is more robust to perturbations in measurement noise variance.

blending parametrization is not suitable for inpainting as reflected by the large gap in FID. To see this, we point out that in *Dirac* $t$ directly parametrizes the severity of the degradation, that is our model learns a continuum of reconstruction problems with smoothly changing difficulty. On the other hand, blending missing pixels with the clean image does not offer a smooth transition in terms of reconstruction difficulty: for any $0 \leq t < 1$ the reconstruction of $x_0$ from $y_t$ becomes trivial. Furthermore, as our model is trained on a wide range of noise levels due to the SDP formulation, we observe improved robustness to test-time perturbations in measurement noise compared to the blending formulation (Fig. 8).

# H  ROBUSTNESS ABLATIONS

**Degradation severity –** We evaluate robustness of *Dirac* against test-time perturbations in the forward process for Gaussian blur. In particular, suppose that the standard deviation of the Gaussian blur kernel is perturbed with a multiplicative factor of $k$ (i.e., $w_{perp} = kw_{max}$). We pick $k \in [0.6, 0.8, 1.0, 1.2, 1.4]$ and plot the change in distortion (SSIM) and perception (LPIPS) metrics on the FFHQ test split (see Figure 9) using our perception-optimized model. We observe that, as is the case for other supervised methods, reconstruction performance degrades (in terms of both distortion and perception metrics) when the degradation model is significantly changed. Nevertheless, we observe that the performance of *Dirac* is almost unchanged under blur kernel standard deviation reductions of up to 20%, which is a significant perturbation. We hypothesize that the robustness of *Dirac* to forward model shifts is due to fact that the model is trained on a range of degradation severities in the input. Furthermore, we observe that distortion metrics, such as SSIM, degrade less gracefully in the increased severity direction, while perception metrics, such as LPIPS, behave in the opposite manner. We expect our distortion optimized model to be more robust in terms of distortion metric degradation when the forward model is perturbed.

**Measurement noise–** We test the robustness of *Dirac* against perturbations of measurement noise variance compared to the training setup. We evaluate our perception-optimized model, trained under measurement noise with $\sigma = 0.05$, on the FFHQ test split on the gaussian deblurring task with measurement noise standard deviations in $\sigma = [0.0, 0.02, 0.04, 0.05, 0.06, 0.08]$. Our model demonstrates great performance when the nosie level is decreased with improved performance in terms of LPIPS compared to the training setting (see Figure 8). For higher noise variances, the performance of *Dirac* degrades more gracefully than other similar techniques such as InDI Delbracio & Milanfar (2023) (see more discussion in Appendix G).

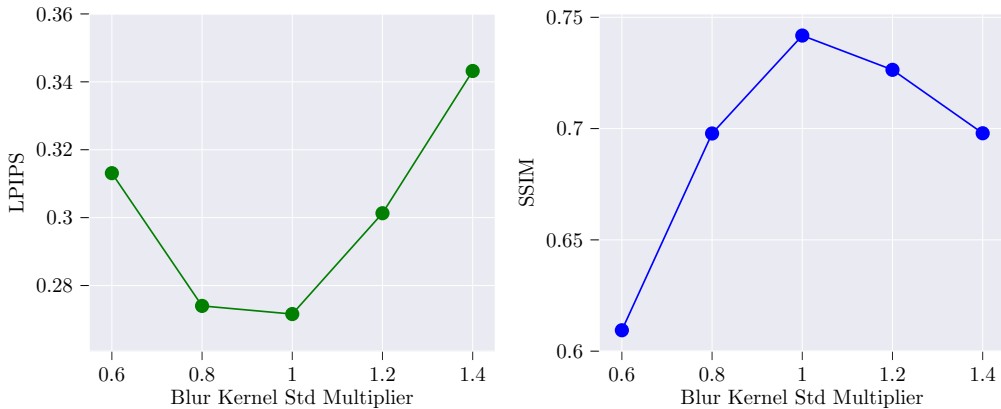

Figure 9: Effect of Gaussian blur kernel width perturbation on the FFHQ test set for the deblurring task. The change in the LPIPS metric (left) together with the SSIM metric (right) is shown.

## I INCREMENTAL RECONSTRUCTION LOSS ABLATIONS

We propose the incremental reconstruction loss, that combines learning to denoise and reconstruct simultaneously in the form

$$\mathcal{L}_{IR}(\Delta t, \boldsymbol{\theta}) = \mathbb{E}_{t,(\boldsymbol{x}_0, \boldsymbol{y}_t)} \left[ w(t) \left\| \mathcal{A}_\tau(\Phi_{\boldsymbol{\theta}}(\boldsymbol{y}_t, t)) - \mathcal{A}_\tau(\boldsymbol{x}_0) \right\|^2 \right], \tag{24}$$

where $\tau = \max(t - \Delta t, 0)$, $t \sim U[0,1]$, $(\boldsymbol{x}_0, \boldsymbol{y}_t) \sim q_0(\boldsymbol{x}_0)q_t(\boldsymbol{y}_t|\boldsymbol{x}_0)$. This loss directly improves incremental reconstruction by encouraging $\mathcal{A}_{t-\Delta t}(\Phi_{\boldsymbol{\theta}}(\boldsymbol{y}_t, t)) \approx \mathcal{A}_{t-\Delta t}(\boldsymbol{x}_0)$. We show in Proposition A.6 that $\mathcal{L}_{IR}(\Delta t, \boldsymbol{\theta})$ is an upper bound to the denoising score-matching objective $\mathcal{L}(\boldsymbol{\theta})$. Furthermore, we show that given enough model capacity, minimizing $\mathcal{L}_{IR}(\Delta t, \boldsymbol{\theta})$ also minimizes $\mathcal{L}(\boldsymbol{\theta})$. However, if the model capacity is limited compared to the difficulty of the task, we expect a trade-off between incremental reconstruction accuracy and score accuracy. This trade-off might not be favorable in tasks where incremental reconstruction is accurate enough due to the smoothness properties of the degradation (see Theorem 3.4). Here, we perform further ablation studies to investigate the effect of the *look-ahead* parameter $\Delta t$ in the incremental reconstruction loss.

**Deblurring –** In case of deblurring, we did not find a significant difference in perceptual quality with different $\Delta t$ settings. Our results on the CelebA-HQ validation set can be seen in Figure 10 (left). We observe that using $\Delta t = 0$ (that is optimizing $\mathcal{L}(\boldsymbol{\theta})$) yields slightly better reconstructions (difference in the third digit of LPIPS) than optimizing with $\Delta t = 1$, that is minimizing

$$\mathcal{L}_{IR}(\Delta t = 1, \boldsymbol{\theta}) := \mathcal{L}_{IR}^{\mathcal{X}_0}(\boldsymbol{\theta}) = \mathbb{E}_{t,(\boldsymbol{x}_0, \boldsymbol{y}_t)} \left[ w(t) \left\| \Phi_{\boldsymbol{\theta}}(\boldsymbol{y}_t, t) - \boldsymbol{x}_0 \right\|^2 \right]. \tag{25}$$

This loss encourages one-shot reconstruction and denoising from any degradation severity, intuitively the most challenging task to learn. We hypothesize, that the blur degradation used in our experiments is smooth enough, and thus the incremental reconstruction as per Theorem 3.4 is accurate. Therefore, we do not need to trade off score approximation accuracy for better incremental reconstruction.

**Inpainting –** We observe very different characteristics in case of inpainting. In fact, using the vanilla score-matching loss $\mathcal{L}(\boldsymbol{\theta})$, which is equivalent to $\mathcal{L}_{IR}(\Delta t, \boldsymbol{\theta})$ with $\Delta t = 0$, we are unable to learn a meaningful inpainting model. As we increase the look-ahead $\Delta t$, reconstructions consistently improve. We obtain the best results in terms of FID when minimizing $\mathcal{L}_{IR}^{\mathcal{X}_0}(\boldsymbol{\theta})$. Our results are summarized in Figure 10 (middle). We hypothesize that due to rapid changes in the inpainting operator, our incremental reconstruction estimator produces very high errors when trained on $\mathcal{L}(\boldsymbol{\theta})$ (see Theorem 3.4). Therefore, in this scenario improving incremental reconstruction at the expense of score accuracy is beneficial. Figure 10 (right) demonstrates how reconstructions visually change as we increase the look-ahead $\Delta t$. With $\Delta t = 0$, the reverse process misses the clean image manifold completely. As we increase $\Delta t$, reconstruction quality visually improves, but the generated images often have features inconsistent with natural images in the training set. We obtain high quality, detailed reconstructions for $\Delta t = 1$ when minimizing $\mathcal{L}_{IR}^{\mathcal{X}_0}(\boldsymbol{\theta})$.

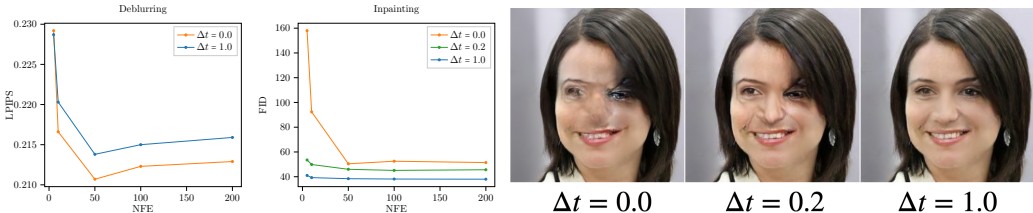

$\Delta t = 0.0$       $\Delta t = 0.2$       $\Delta t = 1.0$

Figure 10: Effect of incremental reconstruction loss step size on the CelebA-HQ validation set for deblurring (left) and inpainting (middle). Visual comparison of inpainted samples is shown on the right.

## J  FURTHER INCREMENTAL RECONSTRUCTION APPROXIMATIONS

In this work, we focused on estimating the incremental reconstruction

$$\mathcal{R}(t, \Delta t; \boldsymbol{x}_0) := \mathcal{A}_{t-\Delta t}(\boldsymbol{x}_0) - \mathcal{A}_t(\boldsymbol{x}_0) \tag{26}$$

in the form

$$\hat{\mathcal{R}}(t, \Delta t; \boldsymbol{y}_t) = \mathcal{A}_{t-\Delta t}(\Phi_{\boldsymbol{\theta}}(\boldsymbol{y}_t, t)) - \mathcal{A}_t(\Phi_{\boldsymbol{\theta}}(\boldsymbol{y}_t, t)), \tag{27}$$

which we call the *look-ahead method*. The challenge with this formulation is that we use $\boldsymbol{y}_t$ with degradation severity $t$ to predict $\mathcal{A}_{t-\Delta t}(\boldsymbol{x}_0)$ with less severe degradation $t - \Delta t$. That is, as we discussed in the paper $\Phi_{\boldsymbol{\theta}}(\boldsymbol{y}_t, t)$ does not only need to denoise images with arbitrary degradation severity, but also has to be able to perform incremental reconstruction, which we address with the incremental reconstruction loss. However, other methods of approximating (26) are also possible, with different trade-offs. The key idea is to use different methods to estimate the gradient of $\mathcal{A}_t(\boldsymbol{x}_0)$ with respect to the degradation severity, followed by first-order Taylor expansion to estimate $\mathcal{A}_{t-\Delta t}(\boldsymbol{x}_0)$.

**Small look-ahead (SLA) –** We use the approximation

$$\mathcal{A}_{t-\Delta t}(\boldsymbol{x}_0) - \mathcal{A}_t(\boldsymbol{x}_0) \approx \Delta t \cdot \frac{\mathcal{A}_{t-\delta t}(\boldsymbol{x}_0) - \mathcal{A}_t(\boldsymbol{x}_0)}{\delta t}, \tag{28}$$

where $0 < \delta t < \Delta t$ to obtain

$$\hat{\mathcal{R}}^{SLA}(t, \Delta t; \boldsymbol{y}_t) = \Delta t \cdot \frac{\mathcal{A}_{t-\delta t}(\Phi_{\boldsymbol{\theta}}(\boldsymbol{y}_t, t)) - \mathcal{A}_t(\Phi_{\boldsymbol{\theta}}(\boldsymbol{y}_t, t))}{\delta t}. \tag{29}$$

The potential benefit of this method is that $\mathcal{A}_{t-\delta t}(\Phi_{\boldsymbol{\theta}}(\boldsymbol{y}_t, t))$ may approximate $\mathcal{A}_{t-\delta t}(\boldsymbol{x}_0)$ much more accurately than $\mathcal{A}_{t-\Delta t}(\Phi_{\boldsymbol{\theta}}(\boldsymbol{y}_t, t))$ can approximate $\mathcal{A}_{t-\Delta t}(\boldsymbol{x}_0)$, since $t-\delta t$ is closer in severity to $t$ than $t - \Delta t$. However, depending on the sharpness of $\mathcal{A}_t$, the first-order Taylor approximation may accumulate large error.

**Look-back (LB) –** We use the approximation

$$\mathcal{A}_{t-\Delta t}(\boldsymbol{x}_0) - \mathcal{A}_t(\boldsymbol{x}_0) \approx \mathcal{A}_t(\boldsymbol{x}_0) - \mathcal{A}_{t+\Delta t}(\boldsymbol{x}_0), \tag{30}$$

that is we predict the incremental reconstruction based on the most recent change in image degradation. Plugging in our model yields

$$\hat{\mathcal{R}}^{LB}(t, \Delta t; \boldsymbol{y}_t) = \mathcal{A}_t(\Phi_{\boldsymbol{\theta}}(\boldsymbol{y}_t, t)) - \mathcal{A}_{t+\Delta t}(\Phi_{\boldsymbol{\theta}}(\boldsymbol{y}_t, t)). \tag{31}$$

The clear advantage of this formulation over (27) is that if the loss in (10) is minimized such that $\mathcal{A}_t(\Phi_{\boldsymbol{\theta}}(\boldsymbol{y}_t, t)) = \mathcal{A}_t(\boldsymbol{x}_0)$, then we also have

$$\mathcal{A}_{t+\Delta t}(\Phi_{\boldsymbol{\theta}}(\boldsymbol{y}_t, t)) = \mathcal{G}_{t \to t+\Delta t}(\mathcal{A}_t(\Phi_{\boldsymbol{\theta}}(\boldsymbol{y}_t, t))) = \mathcal{G}_{t \to t+\Delta t}(\mathcal{A}_t(\boldsymbol{x}_0)) = \mathcal{A}_{t+\Delta t}(\boldsymbol{x}_0).$$

However, this method may also accumulate large error if $\mathcal{A}_t$ changes rapidly close to $t$.

**Small look-back (SLB)–** Combining the idea in SLA with LB yields the approximation

$$\mathcal{A}_{t-\Delta t}(\boldsymbol{x}_0) - \mathcal{A}_t(\boldsymbol{x}_0) \approx \Delta t \cdot \frac{\mathcal{A}_t(\boldsymbol{x}_0) - \mathcal{A}_{t+\delta t}(\boldsymbol{x}_0)}{\delta t}, \tag{32}$$

where $0 < \delta t < \Delta t$. Using our model, the estimator of the incremental reconstruction takes the form

$$\hat{\mathcal{R}}^{SLB}(t, \Delta t; \boldsymbol{y}_t) = \Delta t \cdot \frac{\mathcal{A}_t(\Phi_{\boldsymbol{\theta}}(\boldsymbol{y}_t, t)) - \mathcal{A}_{t+\delta t}(\Phi_{\boldsymbol{\theta}}(\boldsymbol{y}_t, t))}{\delta t}. \tag{33}$$

Compared with LB, we still have $\mathcal{A}_{t+\delta t}(\Phi_{\boldsymbol{\theta}}(\boldsymbol{y}_t, t)) = \mathcal{A}_{t+\delta t}(\boldsymbol{x}_0)$ and the error due to first-order Taylor-approximation is reduced, however potentially higher than in case of SLA.

**Incremental Reconstruction Network –** Finally, an additional model $\phi_{\boldsymbol{\theta}'}$ can be trained to directly approximate the incremental reconstruction, that is $\phi_{\boldsymbol{\theta}'}(\boldsymbol{y}_t, t) \approx \mathcal{R}(t, \Delta t; \boldsymbol{x}_0)$. All these approaches are interesting directions for future work.

### J.1 COMPARISON METHODS

For all methods, hyperparameters are tuned based on first 100 images of the folder `"00001"` for FFHQ and tested on the folder `"00000"`. For ImageNet experiments, we use the first samples of the first 100 classes of ImageNet validation split to tune, last samples of each class as the test set.

#### J.1.1 DPS

We use the default value of 1000 NFEs for all tasks. We make no changes to the Gaussian blurring operator in the official source code. For inpainting, we copy our operator and apply it in the image input range $[0, 1]$. The step size $\zeta'$ is tuned via grid search for each task separately based on LPIPS metric. The optimal values are as follows:

1. FFHQ Deblurring: $\zeta' = 3.0$
2. FFHQ Inpainting: $\zeta' = 2.0$
3. ImageNet Deblurring: $\zeta' = 0.3$
4. ImageNet Inpainting: $\zeta' = 3.0$

As a side note, at the time of writing this paper, the official implementation of DPS[2] adds the noise to the measurement after scaling it to the range $[-1, 1]$. For the same noise standard deviation, the effect of the noise is halved as compared to applying in $[0, 1]$ range. To compensate for this discrepancy, we set the noise std in the official code to $\sigma = 0.1$ for all DPS experiments which is the same effective noise level as $\sigma = 0.05$ for our experiments.

#### J.1.2 DDRM

We keep the default settings $\eta_B = 1.0$, $\eta = 0.85$ for all of the experiments and sample for 20 NFEs with DDIM Song et al. (2021a). For the Gaussian deblurring task, the linear operator has been implemented via separable 1D convolutions as described in D.5 of DDRM Kawar et al. (2022a). We note that for blurring task, the operator is applied to the reflection padded input. For Gaussian inpainting task, we set the left and right singular vectors of the operator to be identity ($\mathbf{U} = \mathbf{V} = \mathbf{I}$) and store the mask values as the singular values of the operator. For both tasks, operators are applied to the image in the $[-1, 1]$ range.

#### J.1.3 PNP-ADMM

We take the implementation from the `scico` library. Specifically the code is modified from the sample notebook[3]. We set the number of ADMM iterations to be `maxiter=12` and tune the ADMM penalty parameter $\rho$ via grid search for each task based on LPIPS metric. The values for each task are as follows:

1. FFHQ Deblurring: $\rho = 0.1$
2. FFHQ Inpainting: $\rho = 0.4$

---

[2]`https://github.com/DPS2022/diffusion-posterior-sampling`
[3]`https://github.com/lanl/scico-data/blob/main/notebooks/superres_ppp_dncnn_admm.ipynb`

3. ImageNet Deblurring: $\rho = 0.1$

4. ImageNet Inpainting: $\rho = 0.4$

The proximal mappings are done via pre-trained DnCNN denoiser with `17M` parameters.

### J.1.4 ADMM-TV

We want to solve the following objective:

$$\arg\min_{\boldsymbol{x}} \frac{1}{2}\|\boldsymbol{y} - \mathcal{A}_1(\boldsymbol{x})\|_2^2 + \lambda\|\mathbf{D}\boldsymbol{x}\|_{2,1}$$

where $\boldsymbol{y}$ is the noisy degraded measurement, $\mathcal{A}_1(\cdot)$ refers to blurring/masking operator and $\mathbf{D}$ is a finite difference operator. $\|\mathbf{D}\boldsymbol{x}\|_{2,1}$ TV regularizes the prediction $\boldsymbol{x}$ and $\lambda$ controls the regularization strength. For a matrix $\mathbf{A} \in \mathbb{R}^{m \times n}$, the matrix norm $\|.\|_{2,1}$ is defined as:

$$\|\mathbf{A}\|_{2,1} = \sum_{i=1}^{m} \sqrt{\sum_{j=1}^{n} \mathbf{A}_{ij}^2}$$

The implementation is taken from `scico` library where the code is based on the sample notebook[4]. We note that for consistency, the blurring operator is applied to the reflection padded input. In addition to the penalty parameter $\rho$, we need to tune the regularization strength $\lambda$ in this problem. We tune the pairs $(\lambda, \rho)$ for each task via grid search based on LPIPS metric. Optimal values are as follows:

1. FFHQ Deblurring: $(\lambda, \rho) = (0.007, 0.8)$

2. FFHQ Inpainting: $(\lambda, \rho) = (0.02, 0.2)$

3. ImageNet Deblurring: $(\lambda, \rho) = (0.007, 0.5)$

4. ImageNet Inpainting: $(\lambda, \rho) = (0.02, 0.2)$

### J.1.5 INDI

In order to ablate the effect of degradation parametrization, we match the experimental setup as closely as possible to *Dirac* setting on CelebA-HQ. We train the same model as used for *Dirac* from scratch. As InDI does not leverage diffusion directly, we train on a weighted $\ell_2$ loss, where $w_t = \frac{1}{t^2 + \epsilon}$ instead of $1/\sigma_t^2$-weighting in our method. We adjust the learning rate to account for the resulting difference in scale. We use our degradation scheduling method from 2 to schedule $t$. For inference, we set $\Delta t = 0.05$.

## K  FURTHER RECONSTRUCTION SAMPLES

Here, we provide more samples from *Dirac* reconstructions on the test split of CelebA-HQ and ImageNet datasets. We visualize the uncertainty of samples via pixelwise standard deviation across $n = 10$ generated samples. In experiments where the distortion peak is achieved via one-shot reconstruction, we omit the uncertainty map.

---

[4]`https://github.com/lanl/scico-data/blob/main/notebooks/deconv_tv_` `padmm.ipynb`

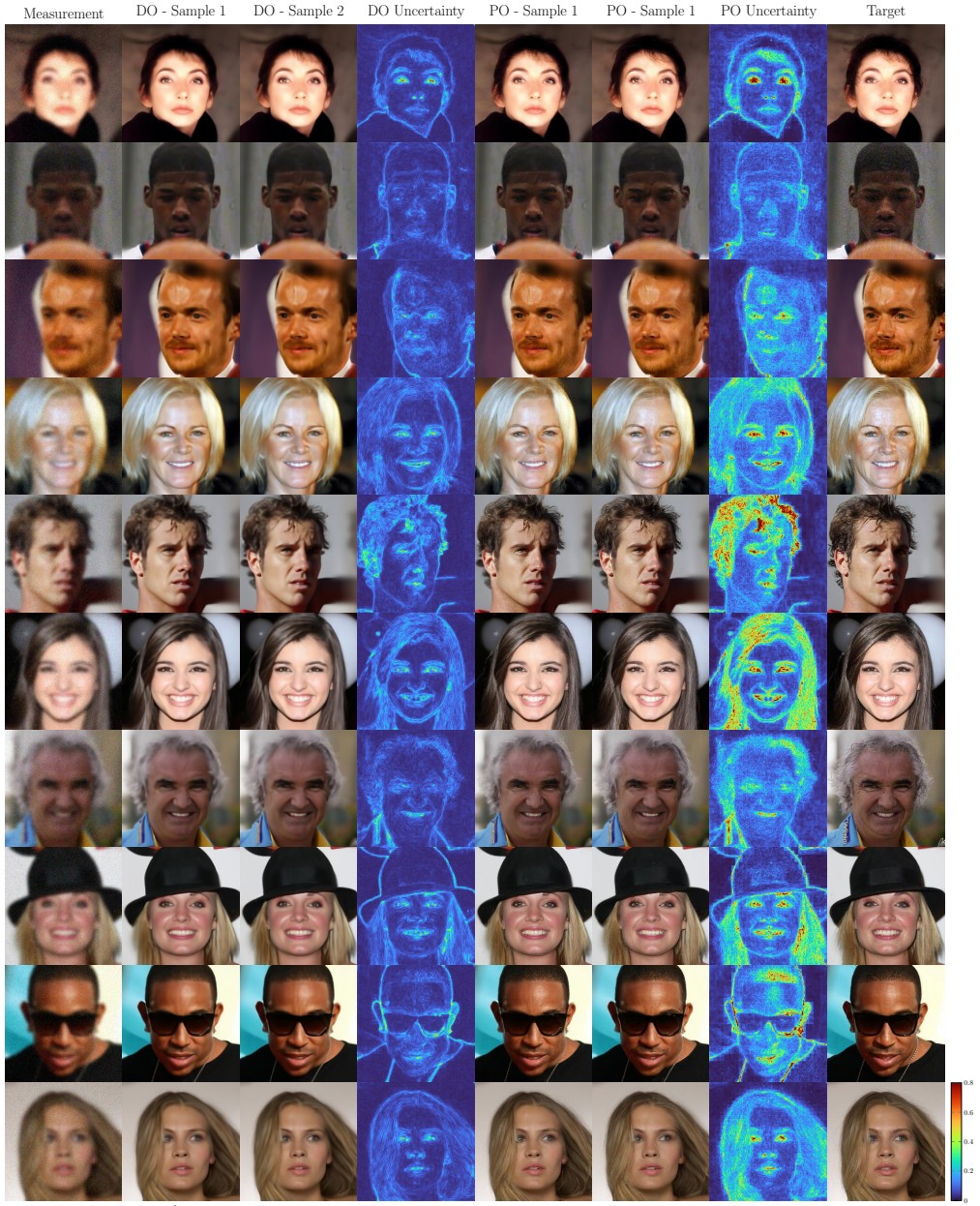

Figure 11: Distortion and Perception optimized deblurring results for the CelebA-HQ dataset (test split). Uncertainty is calculated over $n = 10$ reconstructions from the same measurement.

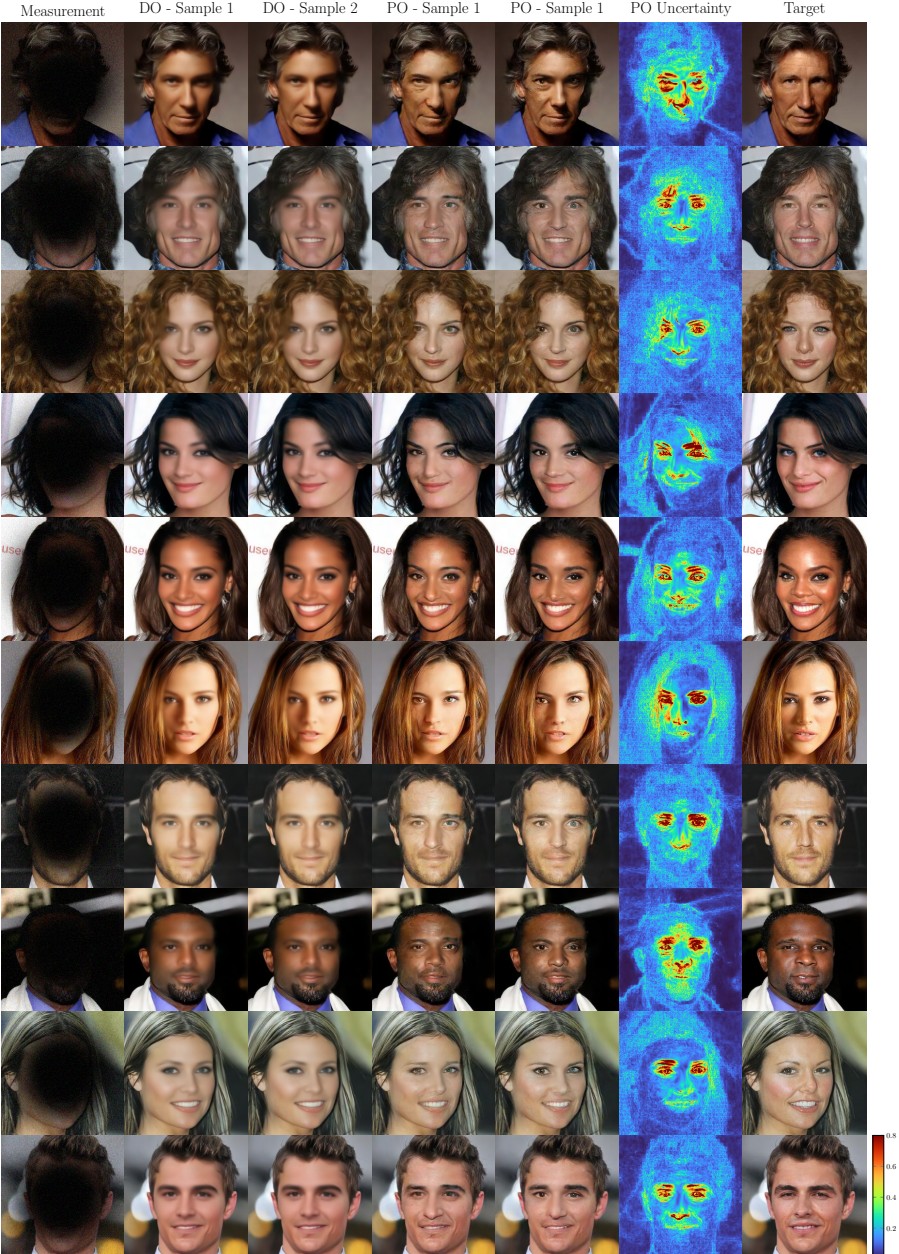

Figure 12: Distortion and Perception optimized inpainting results for the CelebA-HQ dataset (test split). Uncertainty is calculated over $n = 10$ reconstructions from the same measurement. For distortion optimized runs, images are generated in one-shot, hence we don't provide uncertainty maps.

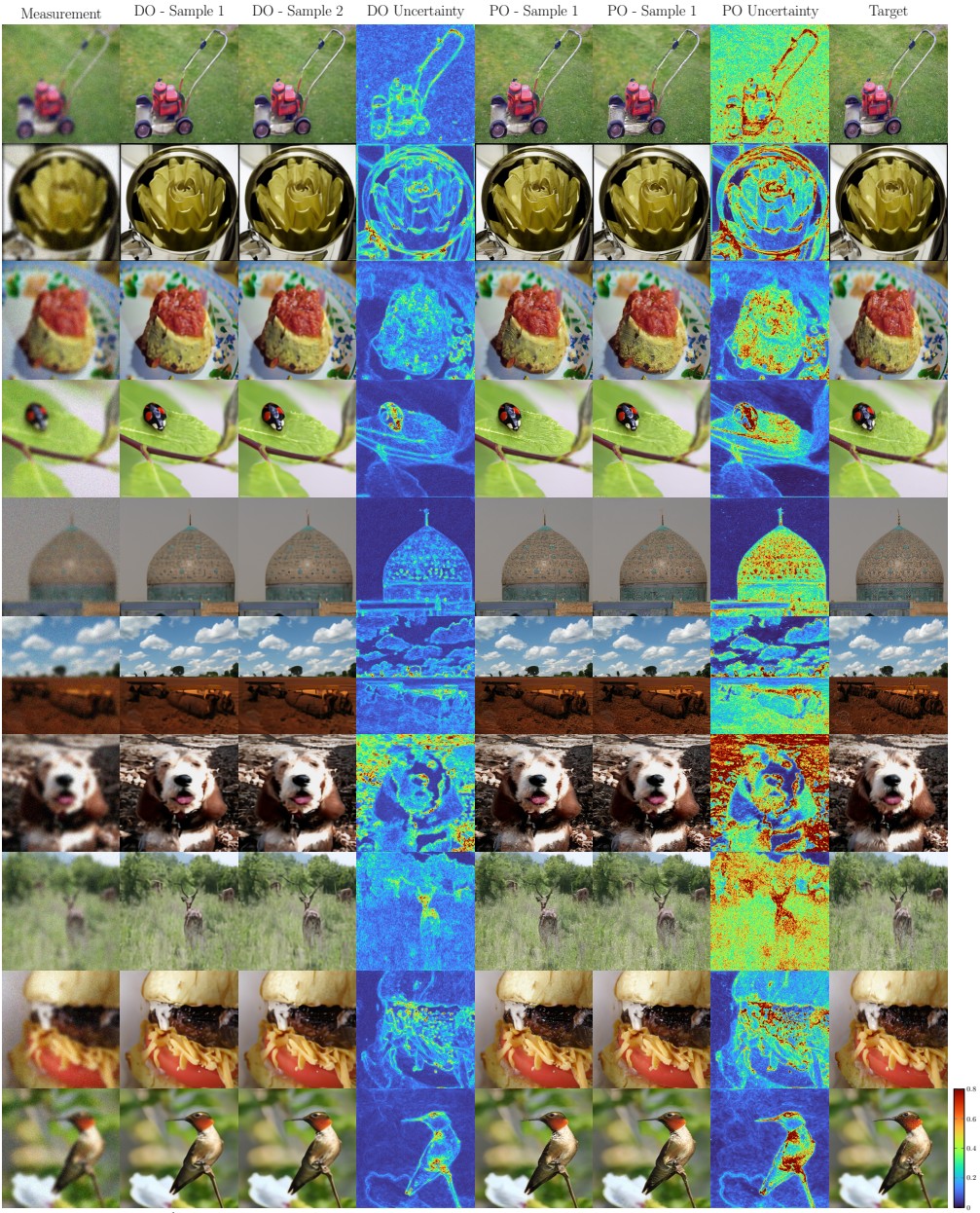

Figure 13: Distortion and Perception optimized deblurring results for the ImageNet dataset (test split). Uncertainty is calculated over $n = 10$ reconstructions from the same measurement.

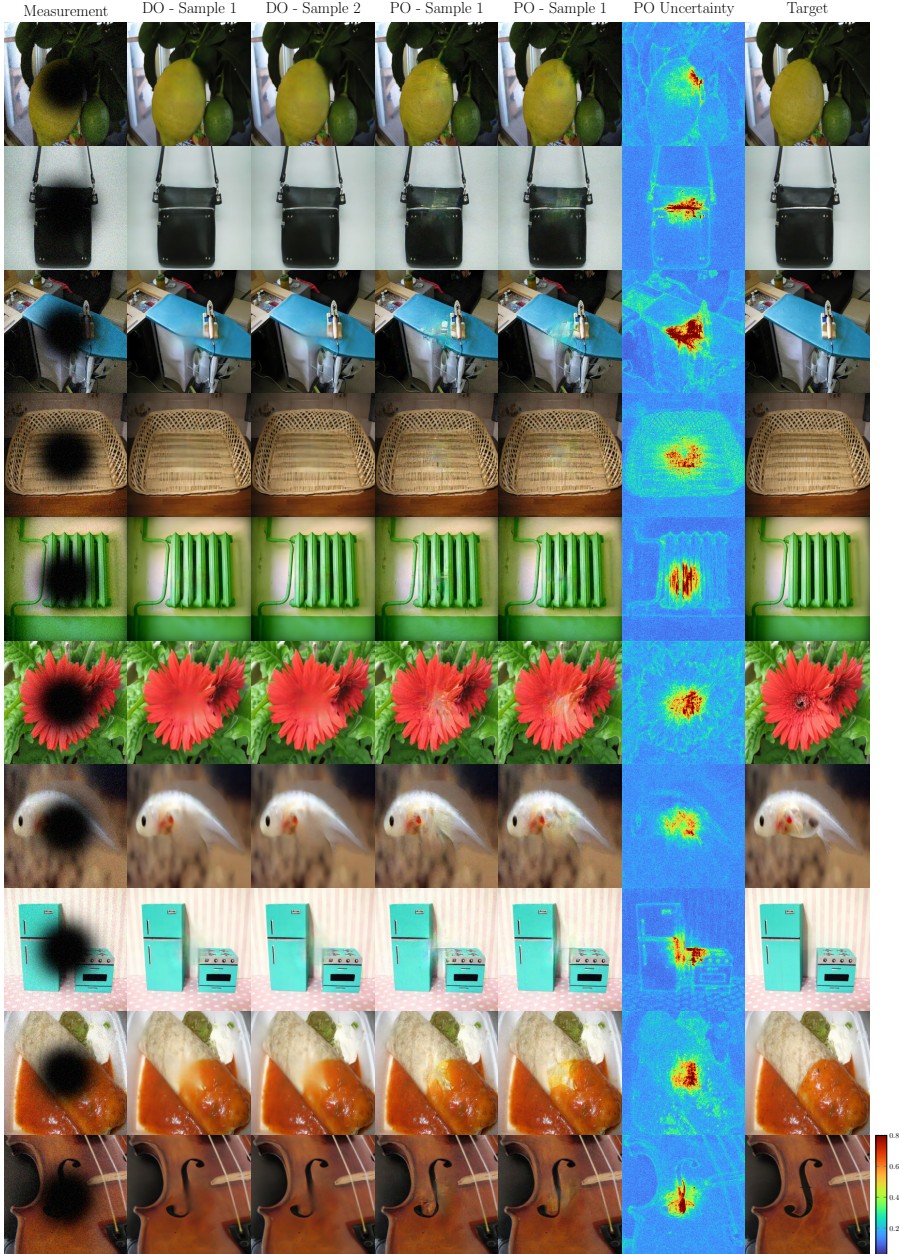

Figure 14: Distortion and Perception optimized inpainting results for the ImageNet dataset (test split). Uncertainty is calculated over $n = 10$ reconstructions from the same measurement. For distortion optimized runs, images are generated in one-shot, hence we don't provide uncertainty maps.

