# OpenReview forum: "DiracDiffusion: Denoising and Incremental Reconstruction with Assured Data-Consistency"
_ICLR.cc/2024/Conference — Submitted to ICLR 2024_

### Official Review · Reviewer_RkZw · 2023-10-19

**Soundness:** 3 good
**Presentation:** 3 good
**Contribution:** 3 good
**Rating:** 6
**Confidence:** 4

**Summary:**

The paper presents a method to solve inverse problems in imaging by means of denoising diffusion models. The main challenge with this lies in enforcing consistency with the forward model of the problem and controlling the distortion-perception tradeoff. The proposed approach maintains consistency with the measurements throughout the reverse diffusion process and can prioritize distortion or perception in the solutions depending on the number of iterations.

**Strengths:**

The paper is generally well-written and the method is sound. The technique used to enforce measurement consistency seems to outperform existing approaches in the same area. Overall, the method is a valuable contribution and has practical relevance for the solution of inverse problems.

**Weaknesses:**

As it is common for most papers on solving inverse problems with diffusion models, the experimental setting is rather limited. Only two degradations (deblurring and inpainting) have been studied so generalization to other ones is unclear (e.g. compressive sensing may exhibit significantly different behaviour). The selection of baselines in the comparion tables is reasonable but I would have added a state-of-the-art method from the supervised training literature (e.g. NAFNet from Chen et al. "Simple Baselines for Image Restoration") as a further reference prioritizing distortion.

**Questions:**

It would be interesting to know what happens under a mismatch of the degradation process between training and testing.

---

> ### Author Response · Authors · 2023-11-15
> **Response to Reviewer RkZw**
>
> Thank you for your time reviewing our manuscript. We are glad to hear that our method is “a valuable contribution” and “has practical relevance for the solution of inverse problems”. Please find below our response.
>
> - Re supervised training baseline: Thank you for your suggestion. We added a state-of-the-art supervised image restoration baseline, SwinIR, to our experiments on the FFHQ dataset. We observe that SwinIR achieves excellent performance in terms of distortion metrics which is closely matched by DiracDiffusion optimized for distortion. However, as expected, DiracDiffusion (and other diffusion-based solvers typically) outperforms supervised non-diffusion restoration methods in terms of perceptual metrics by large margins. We are in the process of running the same experiments on the ImageNet dataset as well, which we are going to add to the final version of the manuscript. However, we expect to see the same trends as in the case of FFHQ.
>
> - Re train-test degradation mismatch: This is a great question. We added a new robustness study with respect to forward model perturbations in test time (see Appendix H). In particular, we perturb the blur kernel width in test time in Gaussian deblurring. We find that, as in other supervised methods, reconstruction performance degrades when the degradation model is significantly changed. However, we observe that performance is almost unchanged under blur kernel standard deviation reductions of up to 20%, which is a significant perturbation. Moreover, the reviewer can find experiments on the noise robustness of DiracDiffusion in Appendix G and H (Figure 8), where we perturb the measurement noise level in test time. We observe that DiracDiffusion is robust with respect to noise level perturbations. We hypothesize that this is due to the fact that our model is trained to denoise on a range of noise variances due to the similarity of our objective to denoising score-matching.
>
> We hope that we have adequately addressed the reviewer’s concerns and we are open for further discussion if needed. We would greatly appreciate it if the reviewer considered updating their rating in case all issues have been addressed, given the reviewer’s generally favorable opinion on our paper.

---

> > ### Author Response · Authors · 2023-11-22
> > **Asking for feedback on the updated version of the manuscript**
> >
> > As the end of the rebuttal period is approaching, we would greatly appreciate it if the reviewer could let us know their opinion on our updated manuscript.  Following the reviewer’s suggestion, we performed additional robustness experiments and added an extra state-of-the-art supervised baseline. Please let us know if you have further concerns and we would like to kindly ask you to update your score reflecting your opinion on the updated manuscript.

---

> > > ### Comment · Reviewer_RkZw · 2023-11-22
> > > **Re: Asking for feedback on the updated version of the manuscript**
> > >
> > > Thanks, the response addressed my questions. I confirm my score.

---

### Official Review · Reviewer_te6k · 2023-10-25

**Soundness:** 3 good
**Presentation:** 3 good
**Contribution:** 3 good
**Rating:** 5
**Confidence:** 3

**Summary:**

This paper is interesting. It proposes a new scheme using assumption that the observation comes from a stochastic degradation process that gradually degrades and noises the original clean image. By learning to reverse the degradation process in order to recover the clean image. The proposed technique maintains consistency with the original measurement throughout the reverse process, and allows for great flexibility in trading off perceptual quality for improved distortion metrics and sampling speedup via early-stopping.
Experiemental results show better trade-offs between both perceptual and distortion metrics.

**Strengths:**

A novel framework by gradually degrading and noises the original clean image, and by learning to reverse the degradation process in order to recover the clean image.
New SOTA results show better trade-offs between both perceptual and distortion metrics.

**Weaknesses:**

Each degradation needs a re-training of the inverse network.
Flexibility to extend to more complex image restoration tasks is unknown.

**Questions:**

Can this network be trained to tackle other types of degradations excluding paper's Guassian denoising and inpainting?

---

> ### Author Response · Authors · 2023-11-15
> **Response to Reviewer te6k**
>
> Thank you for reviewing our paper and we are glad to hear that the reviewer found it interesting and novel. Please find below our response to the reviewer’s concerns.
>
> - Re needs retraining: This is a great point, in fact a limitation of our work is that it needs to be trained for a specific reconstruction task. However, the extra cost in training time pays off greatly in faster inference (we only need 5-20 NFEs for optimal performance vs 1000 in DPS), improved reconstruction fidelity and additional flexibility in trading off distortion for perceptual quality. We would argue that in settings where the degradation model is fixed, our model has clear benefits over other diffusion-based solvers.
>
> - Re other image reconstruction tasks: Thank you for raising this point. In fact, our technique can be used for any image corruption, where the functional form of the degradation is known (superresolution, JPEG compression artifact removal etc) and can be interpolated between the most severe degradation (t=1) and identity (t=0). We hope that the provided experiments sufficiently support the utility of our proposed framework. It would be interesting to further tailor DiracDiffusion for specific popular inverse problems, however this effort is more geared towards future work.
>
> We hope that the reviewer’s concerns have been adequately addressed and we are open to addressing any further questions or concerns. We would greatly appreciate it if the reviewer could update their rating if all issues have been addressed, given that the reviewer assigned generally good scores to our paper.

---

> > ### Author Response · Authors · 2023-11-22
> > **Asking for feedback on the updated manuscript**
> >
> > As the end of the rebuttal period is approaching, we would like to kindly ask the reviewer for their feedback on the final version of our manuscript. We did our best to address all the concerns and we hope we managed to convince the reviewer about the merit of our work. We would be glad to elaborate more if the reviewer has any further concerns and we would greatly appreciate it if you could update your score based on your opinion about the updated version of the manuscript.

---

### Official Review · Reviewer_1HXs · 2023-10-27

**Soundness:** 3 good
**Presentation:** 3 good
**Contribution:** 3 good
**Rating:** 6
**Confidence:** 4

**Summary:**

This paper introduces a framework for solving inverse problems by reversing a known stochastic degradation process. The main advantage of the method is that leads to a more clear compromise in the Perception-Distortion trade off than other works. The core of the method is to define a continuous path of the degradations from the high-quality to the low-quality observation (Definition 3.1). The idea is to reverse the analytical sthocastic degradation with a model trained for this specific purpose. The method is evaluated on a series of inverse problems on popular benchmarks.

**Strengths:**

* The paper addresses the interesting and relevant problem of solving inverse problems using a diffusion strategy.
* The paper is well written and the theoretical presentation is sound and clear.
* Experimental results show that the method performance is competitive to other relevant work.

**Weaknesses:**

* Comparison and discussion to other relevant published work is not complete (see questions below)
* Experimental results compare the proposed method to general frameworks that do not require retraining for specific degradations. In that sense it should be more faire to compare to other methods that train a dedicated model for a given degradation process.

**Questions:**

The paper is in general well written. The two main concerns are related to the two main weknesses described below. That is:

1. **Comparison and discussion to published relevant work**. In particular, I would like the authors to comment how the presented work compares to Daras et al. (2022). It seems that Daras et al. (2022) can be used to solve inverse problems, in a non-blind formulation, similar to what this paper is proposing. Can the same strategy/sampling be used in this context? Additionally, I would like the authors to comment on how the present work compares to Delbracio and Milanfar (2023), since this work also discusses the advantage of their bridge formulation in terms of Perception-Distortion trade off. This work is also similar in the sense that requires to train a dedicated model for a specific degradation, and the convex combination formulation leads to a series of intermediate degraded images. This comment also applies to Bridge methods that are applied to solving inverse problems, as the one in Liu, G.H., et al $I^2$ SB: Image-to-Image Schr\" odinger Bridge, ICML 2023)

2. **Experimental results**. The experimental results should also include methods that require training of finetuning a diffusion prior for solving a specific inverse problems as what the authors are proposing. I wouuld like the authors to comment on this point

[After Rebuttal]
My concerns have been addressed so I'm updating my score to above acceptance threshold.

---

> ### Author Response · Authors · 2023-11-15
> **Response to Reviewer 1HXs**
>
> We thank the reviewer for their effort reviewing our work. We are glad that the reviewer thinks that the problem is “interesting and relevant” and that the paper is “well written and the theoretical presentation is sound and clear”. Please find below our reply to your concerns.
>
> **Re Question 1**: Thank you for bringing up these relevant works.
>
> - Re Daras et al. (2022): We build upon the work in Daras et al. (2022) and propose a novel framework specifically tailored for inverse problem solving and extending the formulation to general corruptions (they consider linear corruptions only). In particular, Soft Diffusion is defined as a solely generative framework, and thus it does not cover sampling from a posterior distribution given noisy measurements. Therefore, the considerations around data consistency, which is the main focus of our work, does not arise in their setting.
>
> - Re Delbracio and Milanfar (2023): In fact, we did compare our method with InDI (see Appendix G for a discussion and experimental results ). Overall, we found that InDI and DiracDiffusion provide similar perceptual reconstruction quality (we achieve slightly better FID, InDI has slightly better LPIPS). However, we have found that DiracDiffusion is more robust with respect to noise level perturbations in test time. We hypothesize that this is due to the fact that our model is trained to denoise on a range of noise variances due to the similarity of our objective to denoising score-matching. However, InDI reconstructs samples from convex combinations between noisy and clean images, and thus the model has not seen any other noise levels but the one used for training. We expect similar observations for Bridge methods as well. We would also like to highlight that these techniques work under a different framework from ours, where the forward model is unknown and thus their focus is not on data consistency (in fact it cannot be evaluated without knowing the forward process).
>
> **Re Question 2:** Thank you for this suggestion. As we mentioned above, we provided a comparison on FFHQ with InDI which is a trained diffusion model-based solver. However, at the time of writing this paper the source code and checkpoints have not been publicly released and thus we had to rely on our own reproduction of the method with computationally heavy training. We are unfortunately unaware of diffusion solvers with fast fine-tuning for adaptation to various inverse problems, but we are happy to compare if the reviewer has some suggestions. Moreover, we added a strong supervised image restoration baseline (SwinIR) to our main result in the newer version of the manuscript that the reviewer might find useful. Based on the feedback, we would be open to moving the discussion on InDI to the main paper in the final version of the manuscript.
>
> Please let us know if you have any further questions or suggestions. We hope that we managed to address the reviewer’s major concerns, in which case we would really appreciate it if the reviewer could update their rating given the generally favorable review.

---

> > ### Comment · Reviewer_1HXs · 2023-11-21
> > **Re Question 1.**
> >
> > I appreciate the author's response to my concerns.
> >
> > Also, I appreciate the authors comparing Delbracio and Milanfar (2023) and Heitz et al (2023) since they are very relevant work (though assuming different things). Nonetheless, in the main body of the manuscript this is the only mention to one of this works:
> >
> > *"Recent work Delbracio & Milanfar (2023) defines the forward process as convex combinations between the clean image and the corrupted observation."*
> >
> > And there is no mention of Heitz et al (2023). Moreover there is no real discussion about these works, nor pointers towards the Appendix (One needs to navigate the Appendix to see what is there, and this shouldn't be the case. The idea of an Appendix is that from the main body we point to the Appendix for specific supplementary material).
> >
> > A second observation is that, the next paragraph states:
> >
> > *"Despite all of the successes of diffusion models in high-quality image generation, the requirements imposed on inverse problems are markedly different from image synthesis. Due to the perception distortion trade-off Blau & Michaeli (2018), as diffusion models generate images of exceptional detail, typically reconstruction methods relying on diffusion underperform in distortion metrics, such as PSNR and SSIM Chung et al. ... "*
> >
> > The related work section requires to include a more fair view of current work addressing image restoration (in particular, notably diffusion works doing image restoration), and also to mention the works that already discuss the Perception -- Distortion tradeoff in this context. In the current exposition it gives the impression that this is the first work addressing explicitly this issue and this is not precise (not sure what's the first work to be honest but at least there should be a discussion of other works doing the same).

---

> > > ### Author Response · Authors · 2023-11-22
> > > **Response to Reviewer 1HXs**
> > >
> > > Thank you for the further feedback. We agree with the reviewer that the comparison with Delbracio and Milanfar (2023) could have been better highlighted and that more discussion on prior work with respect to the perception-distortion trade-off is justified. To address your points we updated the manuscript with the following changes:
> > >
> > > - We included the work of Heitz et al (2023) in the introduction along with Delbracio and Milanfar (2023).
> > >
> > > - We added an extra discussion on the connection between our framework and blending, namely blending can be thought of as a deterministic degradation process conditioned on the degraded image. This can be found in Section 3.2. We also added a pointer to the extra discussion in the Appendix as the reviewer suggested.
> > >
> > > - We added a more thorough overview of prior work observing the perception-distortion trade-off in the context of iterative restoration/diffusion. Furthermore, we restructured the introduction to better highlight related diffusion-based image restoration methods.
> > >
> > > As we mentioned earlier, given an extra page in the camera-ready version we would be open to move the comparison experiment with blending to the main paper. We highlighted the new additions in response to the reviewer’s feedback in blue in the updated manuscript. We hope that the above changes have sufficiently addressed your concerns. Thank you again for your constructive feedback and please let us know if you have further concerns.

---

> > > > ### Comment · Reviewer_1HXs · 2023-11-22
> > > > **Re: Response to Reviewer 1HXs**
> > > >
> > > > Thanks for the response. My concerns have been addressed and therefore I'm updating my score.

---

### Official Review · Reviewer_5bsx · 2023-10-31

**Soundness:** 2 fair
**Presentation:** 3 good
**Contribution:** 2 fair
**Rating:** 5
**Confidence:** 4

**Summary:**

This paper proposes a diffusion model based method for inverse problem solving, with application to computer vision tasks such as image restoration. The authors consider a model that the observation is generated from a stochastic degradation process that gradually degrades the clean image, under which a model is learned to reverse the degradation process to restore the clean image. The proposed method is shown to maintain consistency with the original measurement in the reverse process. Moreover, an early-stopping scheme is adopted for distortion-perception tradeoff. Experimental results on synthetic deblurring and inpainting have been provided to demonstrate the better performance of the proposed method.

**Strengths:**

1. The proposed method is sound and the synthetic experimental results well verified the effectiveness of the proposed method.
2. It is shown the proposed method well maintains data consistency with the original measurement in the reverse process. Moreover, an early-stopping scheme to control the distortion-perception trade-off is proposed.
3. Analysis on the accuracy and limitations of the proposed method have been provided and verified by experimental results.

**Weaknesses:**

1. My main concern is that, while the proposed method is sound, it relies on an assumption that seems unrealistic. Specifically, it assumes the degradation model $A$ being a priori known, which is unrealistic in practical applications, even in the tasks of deblurring and inpainting considered in this paper. The provided experimental results are based on synthetic data, in which the degradation model $A$ can be easily specified. However, when considering practical tasks, I believe the proposed method would not perform satisfactorily, as it is hard or even impossible to accurately specify the degradation model $A$. The authors have not provided evaluation on practical tasks, e.g., practical deblurring and inpainting tasks, in comparison with representative methods.

2. The proposed perception-distortion tradeoff method by early-stopping the reverse process is quite straightforward and heuristic.

3. Experiment details are lacking in Appendix. It seems that the degradation function $A_t(·)$ is necessary in the training process. However, for the deblurring problem, $A(·)$ is unknown, only clean image $x$ and its degraded version $\hat{x}$ are available. How to design $A_t(·)$ with only $x$ and $\hat{x}$?

4. In equation (10), the network learns to reconstruct $A_t(x0)$ rather than directly reconstructing $x_0$ or noise, compared with DDPM. However, when the degradation function $A_t(·)$ is linear, it would be equivalent to reconstructing $x_0$. In some tasks, the degradation model $A(·)$ would be linear, for example, the blur function $y=k*x + n$ in the deblurring task, where $k$ is a blur kernel. How can the deblurring experiment show the superiority of Dirac over DDPM when the degradation model $A(·)$ is linear?

**Questions:**

1. There should be a space in “projectionSong”, line 11, the second paragraph of Introduction.
2. See the Weaknesses above.

---

> ### Author Response · Authors · 2023-11-15
> **Response to Reviewer 5bsx**
>
> Thank you for taking your time reviewing our paper. We are glad that the reviewer thinks that “the proposed method is sound” and that the “experimental results well verified the effectiveness of the proposed method”. Please find below our response to your concerns.
>
> - Re Weakness 1: As the reviewer points out, DiracDiffusion is a solver for inverse problems with *known* forward model. This is a clear distinction from general image restoration techniques that rely only on clean-noisy paired data for training without knowing the corruption process. However, in this general scenario there is no notion of *data consistency*: given a corrupted input and the reconstructed image we are unable to evaluate how well the reconstruction explains the input (without a ground truth). In stark contrast, our work focuses on inverse problems with known forward models, a common case in real-world problems where the physical process producing the observations is known (known camera blur, microscopy with known probe function, medical imaging problems such as CT or MRI etc). In this setting, a crucial requirement is data consistency: the reconstruction has to be consistent with the noisy measurement as dictated by the forward model. In fact, ensuring this consistency while producing exceptional image fidelity is the main goal of our paper. We would like to also highlight that other diffusion-based solvers overwhelmingly assume a known forward model as well ([1-3]). To address the reviewer’s concern we updated the main text to better emphasize this assumption.
>
> - Re Weakness 2: Our early stopping method is indeed straightforward and intuitive, yet very effective. Furthermore, the early stopping technique is an interesting consequence of our proposed Stochastic Diffusion Process framework, and not the sole contribution of our work.
>
> - Re Weakness 3: Thank you for pointing this out, we added the specific form of the degradation process in Appendix F. We emphasize that $\mathcal{A}_t(\cdot)$ is designed a priori based on the known forward model and its time-dependence (scheduling) is determined via the degradation scheduling algorithm in Algorithm 2 in the appendix.
>
> - Re Weakness 4: We point out that when the degradation process is linear ($\mathcal{A}_t(\cdot)$ is a linear mapping), reconstructing $\mathcal{A}_t(x_0)$ is not equivalent to reconstructing $x_0$ as the linear mapping is typically not invertible (as is the case in deblurring or inpainting). In fact, as we highlight in the paper, predicting $\mathcal{A}_t(x_0)$ from the noisy and degraded input (denoising only) is an easier problem than completely reconstructing $x_0$ in one step (denoising and reconstruction/inversion).
>
> Thank you for catching the typo, we fixed it in the new version. We are happy to discuss further if there are additional concerns and would like to ask the reviewer to consider raising their score in case we addressed all major issues.
>
> [1] Kawar, Bahjat, et al. "Denoising diffusion restoration models." Advances in Neural Information Processing Systems 35 (2022): 23593-23606.
>
> [2] Chung, Hyungjin, et al. "Diffusion posterior sampling for general noisy inverse problems." arXiv preprint arXiv:2209.14687 (2022).
>
> [3] Song, Yang, et al. "Solving inverse problems in medical imaging with score-based generative models." arXiv preprint arXiv:2111.08005 (2021).

---

> > ### Author Response · Authors · 2023-11-22
> > **Asking for feedback on the updated version of the manuscript**
> >
> > As the end of the rebuttal period is approaching, we would like to kindly ask the reviewer for their feedback on the updated version of our manuscript. We did our best to address all the concerns of the reviewer. We would greatly appreciate it if the reviewer could let us know about any further concerns and update their scores based on their opinion about the final version of the manuscript.

---

### Author Response · Authors · 2023-11-15
**Summary of updates to the manuscript**

We thank all reviewers for spending time reviewing our paper. We found the reception of our work overall favorable, and we did our best to address all concerns raised by the reviewers. We have also incorporated much of the feedback into the manuscript, with the new additions highlighted in red. The updates are as follows:

- Following the suggestion of **Reviewer 1HXs** and **Reviewer RkZw**, we added experimental results on a strong, state-of-the-art supervised image restoration baseline, SwinIR, to the main paper. The new results can be found in Table 1.

- Following the suggestion of **Reviewer RkZw**, we added a new robustness study in the appendix on test-time forward model perturbation. We investigate both noise level and degradation severity perturbations in Appendix H.

- Addressing the issues raised by **Reviewer 5bsx**, we clarified the exact form of the degradation process used in the experiments in Appendix F, and highlighted in the paper that the forward model needs to be known for DiracDiffusion.

We believe that the above additions further strengthened our paper. Moreover, we are open for further active discussion with the reviewers during the rebuttal period.

---

### Meta-Review · Area_Chair_MNtj · 2023-12-13

**Metareview:**

This paper presents a diffusion model aimed at inverse problem solving, specifically for image restoration tasks. The authors propose a model that progressively degrades a clean image and learns to reverse this process, maintaining data consistency and enabling control over the distortion-perception tradeoff. The method is tested using synthetic examples of deblurring and inpainting and employs early stopping to enhance performance.

 (a) Strengths of the paper

The paper is recognized for its solid theoretical approach and successful synthetic experimental outcomes, which effectively demonstrate the capabilities of the proposed method. The early-stopping scheme to control the distortion-perception trade-off is also noted as a practical contribution to the field. The strengths in methodology and experimental validation are widely acknowledged (Reviewers 5bsx, 1HXs, te6k, and RkZw).

 (b) Weaknesses of the paper

Common weaknesses include concerns about the model's practical applicability and generalization to real-world tasks due to its dependence on a known degradation process, which is not always realistic or available in practical scenarios. Additionally, there is a notable absence of extensive experimental results on real-world applications and a comprehensive comparison to a broader range of existing methods. These concerns about the model's practical utility and the scope of its validation are shared by Reviewers 5bsx, 1HXs, and RkZw.

Although the reviewers recognize the strengths of the paper in theoretical development and synthetic experimentation, yet, the noted weaknesses, particularly regarding practical application and comparative analysis, are significant enough to consider a potential rejection. The authors would need to address these gaps to strengthen the paper's contribution to the field.

**Justification For Why Not Higher Score:**

The paper presents an interesting theoretical approach to solving inverse problems with diffusion models, but there are several reasons it does not warrant acceptance. The method assumes a known degradation model, which several reviewers (5bsx, 1HXs, RkZw) have pointed out is often not practical or available in real-world scenarios, limiting the method's applicability. Additionally, the paper lacks a broad comparison with existing methods and does not provide substantial evidence of performance on real-world tasks, which is essential for demonstrating the method's utility and generalization. These significant gaps in practical validation and comparative analysis, emphasized by the reviewers, suggest that the method while promising, is not yet ready for acceptance without further development and empirical testing in more realistic settings.

**Justification For Why Not Lower Score:**

N/A

---

### Decision · Program_Chairs · 2024-01-16

Reject